# Guardians of Image Quality: Benchmarking Defenses Against Adversarial Attacks on Image Quality Metrics

Aleksandr Gushchin [1 2 3]  Khaled Abud [2 3]  Georgii Bychkov [1 2 3]  Ekaterina Shumitskaya [1 2 3]
Anna Chistyakova [1 3]  Sergey Lavrushkin [1 2]  Bader Rasheed [4]  Kirill Malyshev [3]  Dmitriy S. Vatolin [1 2 3]
Anastasia Antsiferova [1 2 4]

## Abstract

Modern neural-network-based Image Quality Assessment (IQA) metrics are vulnerable to adversarial attacks, which can be exploited to manipulate search engine rankings, benchmark results, and content quality assessments, raising concerns about the reliability of IQA metrics in critical applications. This paper presents the first comprehensive study of IQA defense mechanisms in response to adversarial attacks on these metrics to pave the way for safer use of IQA metrics. We systematically evaluated 30 defense strategies, including purification, training-based, and certified methods — and applied 14 adversarial attacks in adaptive and non-adaptive settings to compare these defenses on 9 no-reference IQA metrics. Our proposed benchmark aims to guide the development of IQA defense methods and is open to submissions; the latest results and code are at https://msu-video-group.github.io/adversarial-defenses-for-iqa/.

## 1. Introduction

Image quality assessment (IQA) metrics are essential to develop and evaluate image and video processing algorithms. Modern IQA metrics based on neural networks are highly correlated with subjective assessments. However, neural networks are proven to be vulnerable to adversarial perturbations(Kurakin et al., 2018), which have led to exploration

of such vulnerabilities of IQA models (Antsiferova et al., 2024; Meftah et al., 2023; Zhang et al., 2024; Ghildyal & Liu, 2023). Adversarial attacks on IQA metrics are perturbations that mislead the metric's score, making quality assessment invalid. Such attacks can manipulate image search results, as search engines (e.g., Microsoft's Bing) rely on IQA metrics to rank results (Bing, 2013). Furthermore, since IQA metrics serve in public benchmarks and comparisons (Huang et al. 2024; Wu et al. 2024, etc.) to evaluate image/video processing and compression algorithms, competitors can exploit the vulnerabilities of the metric to artificially inflate the quality of their algorithm. Several works showed that optimizing image restoration for modern vulnerable IQA metrics can reduce actual image quality (Ding et al., 2021) or generate visual artifacts (Kashkarov et al., 2024). For these reasons, it becomes necessary to study and design methods to improve the robustness of IQA models.

Although researchers have proposed various defense methods to enhance the robustness of neural networks in different applications, few defenses have been developed explicitly for IQA metrics, and there are currently no comprehensive benchmarks for this task. Defending IQA metrics poses unique challenges compared to object classification: defenses must restore the original IQA scores and their correlation with subjective evaluations while preserving the perceptual quality of attacked images.

This paper introduces the first benchmark that systematically evaluates defenses against adversarial attacks on IQA metrics. The evaluation scheme is presented in Figure 1. Our contributions include a novel methodology for measuring and comparing defenses for the IQA tasks, addressing a critical gap in the field, comprehensive experiments, an extensive subjective study with 60,000+ responses, an in-depth analysis of the results, and an online leaderboard. We also publish a novel dataset of adversarial images, which can be used for adversarial training, evaluating non-adaptive defenses, and training methods for attack detection. Our methodology is the first to systematize defenses for IQA metrics, analyzing 30 defense algorithms (both empirical

[1]ISP RAS Research Center for Trusted Artificial Intelligence, Moscow, Russia [2]MSU Institute for Artificial Intelligence, Moscow, Russia [3]Lomonosov Moscow State University, Moscow, Russia [4]Innopolis University, Innopolis, Russia. Correspondence to: Aleksandr Gushchin <alexander.gushchin@graphics.cs.msu.ru>, Anastasia Antsiferova <aantsiferova@graphics.cs.msu.ru>.

*Proceedings of the 42^nd International Conference on Machine Learning*, Vancouver, Canada. PMLR 267, 2025. Copyright 2025 by the author(s).

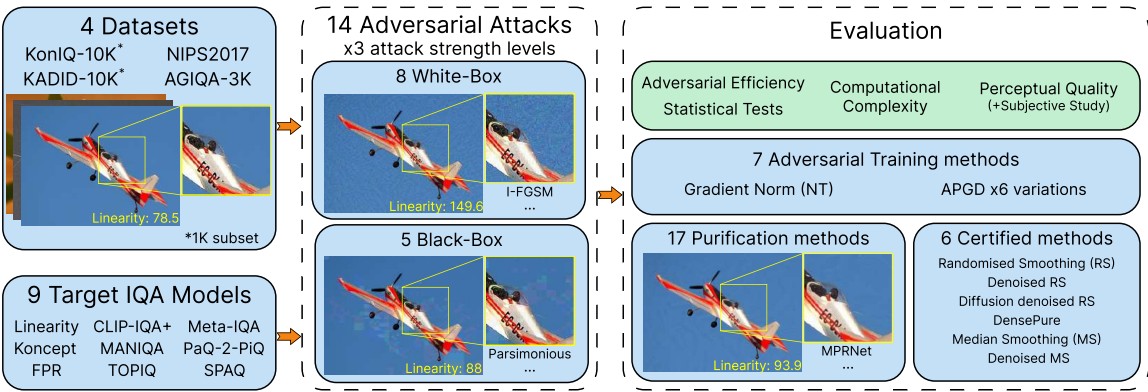

*Figure 1.* **Evaluation scheme**. The benchmark consists of four parts: datasets, IQA models, adversarial attacks, and adversarial defenses of three types: Adversarial Training, Purification and Certified methods.

and certified) and evaluating their effectiveness against 14 adversarial attacks. We address both adaptive and non-adaptive attack scenarios, depending on the attacker's awareness of the defense. The benchmark is available online at https://videoprocessing.ai/benchmarks/iqa-defenses.html along with the code for the proposed methodology, IQA models, adversarial attacks, and defense methods in the GitHub repository. This unified framework enables researchers to measure and compare defense performance, and we welcome submissions of new methods.

## 2. Related work

Existing comparisons of defense methods mostly focus on object classification (Croce et al., 2021; Dong et al., 2020), leaving a significant gap in evaluating defenses for image quality assessment (IQA) metrics. While (Antsiferova et al., 2024) investigates IQA robustness under adversarial attacks, it does not explore defense strategies. Current trends in IQA metrics development emphasize creating task-specific metrics to achieve better correlations, as different IQA tasks (e.g., user- or AI-generated content, artificial distortions caused by image-processing algorithms, etc.) require a slightly different approach. This makes the development of a universally efficient and robust IQA metric impractical. To address this, we present a systematic comparison of defense methods for IQA tasks, enabling researchers to enhance existing models. Our benchmark advances prior work by incorporating attack parameter selection, diverse datasets, and large-scale subjective evaluations, offering a comprehensive framework for assessing defense efficacy.

Adversarial attacks fall into two main categories depending on the attacker's knowledge of the model: "white box" or "black box" (Huang et al., 2017). White-box attacks employ intrinsic characteristics of the attacked models (e.g. gradients); however, in some situations, it is unavailable,

and black-box attacks remain applicable. Several white-box adversarial attacks (Shumitskaya et al., 2024; Zhang et al., 2022b; Wang & Simoncelli, 2008; Shumitskaya et al., 2023) and at least two black-box attacks (Ran et al., 2024; Yang et al., 2024) are designed specifically for IQA metrics.

Defense methods for neural networks come in certified and empirical types. Certified methods provide deterministic or probabilistic robustness guarantees for particular perturbations, datasets, or model architectures. However, these methods are usually computationally complex and reduce the model's general accuracy. One of the most well-known certified methods is randomized smoothing (Cohen et al., 2019). Later variations appeared in (Salman et al., 2020; Chen et al., 2022b), and included a denoiser to improve the defended model's performance. Empirical methods lack robustness guarantees but require fewer computational resources. A widely used empirical defense method is adversarial training (AT) (Wong et al., 2020; Singh et al., 2023), which additionally trains the model on adversarial examples. Vanilla adversarial training, however, may decrease model performance. Adjusting subjective scores during training has been suggested to prevent performance degradation (Chistyakova et al., 2024). In another study, $l_1$-regularization of the gradient norm (NT) was used to improve the robustness of NR IQA models against adversarial attacks (Liu et al., 2024). Adversarial purification is an empirical method that removes adversarial perturbations by processing input data. Although adversarial purification is model-agnostic and computationally efficient, it may fail to eliminate advanced adversarial perturbations and can degrade image quality. Examples of such methods range from compression and spatial transformations to specialized methods (e.g. DiffPure (Nie et al., 2022)).

# 3. Methodology

## 3.1. Problem definition

**Adversarial attacks**. This work evaluates adversarial defenses for no-reference (NR) IQA metrics because they have a more comprehensive range of applications and are more vulnerable to attacks (Ghildyal & Liu, 2023). In this setting, an attacked model, represented by an NR IQA metric, takes a single image as input and estimates image quality. Formally, the NR IQA metric is the mapping $f_\omega : X \rightarrow \mathbb{R}$, parameterized by the vector of weights $\omega$. Here, $X \in [0, 1]^{3 \times H \times W}$ is the input image. An adversarial attack $A : X \rightarrow X$ is the perturbation of the input image defined as

$$A(x) = \underset{x':\rho(x',x)\leq\varepsilon}{\arg\max} \ L(f_\omega(x')), \qquad (1)$$

where $L$ is a loss function that represents the model's outputs for perturbed images and $\rho(\cdot, \cdot)$ is the distance function defined on $X \times X$. We increase IQA scores during the attack to reflect real-life applications (decreasing scores is nearly identical task (Antsiferova et al., 2024)). For IQA metric attacks, we define $L(f_\omega(x')) = \frac{f_\omega(x')}{\text{diam}(f_\omega)}$, where $\text{diam}(f_\omega) = \underset{x,z\in X}{\sup} \{|f_\omega(x) - f_\omega(z)|\}$ represents the range of IQA metric values.

**Adversarial defenses**.

*Adversarial purification* is an algorithm $P : X \rightarrow X$ that aims to transform the input image according to the following optimization problem:

$$\min |f_\omega(P(x')) - f_\omega(x)| + \lambda\rho(P(x'), x), \qquad (2)$$

where $x'$ is the adversarial image, $\lambda$ controls regularization.

*Adversarial training* is formulated as the following problem:

$$\min_\omega \mathbb{E}_{(x,y)\sim\mathcal{D}} \left[ \max_{\|\delta\|_p\leq\varepsilon} \mathcal{L}(f_\omega(x + \delta), y) \right], \qquad (3)$$

where $\mathcal{D}$ is the distribution of training data, $\mathcal{L}$ is a training loss function, $\varepsilon$ is the attack magnitude, $y$ is image quality of $x$. In practice, adversarial training uses an adversarial attack rather than internal maximization.

*Certified methods* used in our paper are based on randomized smoothing (Cohen et al., 2019), denoised randomized smoothing (Salman et al., 2020), diffusion-based randomized smoothing (Carlini et al., 2022; Chen et al., 2022b) and median smoothing (Chiang et al., 2020). Randomized smoothing replaces the original IQA metric $f_\omega(x)$ with a smoothed version $g(x)$, adding Gaussian noise $\epsilon$:

$$g(x) = \underset{\epsilon\sim\mathcal{N}(0,\sigma^2)}{\mathbb{E}} f_\omega(x + \epsilon) \qquad (4)$$

## 3.2. Adversarial attacks

Accurately evaluating defense mechanisms requires testing under conditions that closely reflect real-world applications. To address this, we consider both adversarial attack scenarios: non-adaptive and adaptive. In the first case, the attack method targets only the IQA model itself and does not take the defense into account. In the second, we incorporate differentiable defense into the attacked IQA metric, enabling adaptive attacks to leverage gradients from both the metric and the defense mechanism. We selected 14 white- and black-box attacks of diverse types, *including methods tailored specifically for IQA task*. Table 7 in the Appendix describes these attacks.

**Attacks hyperparameters**. Recent work (Dong et al., 2020) reveals that defense rankings are sensitive to attack parameters. This instability underscores the necessity of evaluating defenses across diverse attack configurations. To account for this, we execute each attack method with three hyperparameter sets corresponding to "weak", "medium", and "strong" perturbation budget. Parameter selection was performed by linear approximation for the target attack budgets on a small subset. We chose a subset of 50 images used for attack alignment via clustering the KonIQ-10k dataset by spatial complexity (SI), colorfulness, and ground-truth quality (MOS). Appendix A.3.2 contains a list of chosen parameter sets, alongside the procedure scheme in Fig.7.

## 3.3. Adversarial defenses

To thoroughly investigate the effectiveness of IQA metric defenses, we explored three method types: adversarial purification, adversarial training, and certified robustness.

**Adversarial purification** mitigates threats by preprocessing input images before IQA calculation. These methods are efficient and offer a flexible trade-off between attack mitigation and image quality. The top part of Table 1 describes the selected adversarial purification techniques. We used five parameter sets to vary the defense strength, e.g., scaling ratio, blurring kernel size, and number of diffusion steps. The Appendix A.3.5 provides a list of used defense parameters and their selection methodology.

**Adversarial training** fine-tunes a model on adversarial examples to enhance its robustness. IQA presents additional challenges in applying adversarial training since adversarial examples don't preserve ground truth labels (MOS). Manually assigning such images with subjective scores is impractical, and using ground-truth labels from clean images is inaccurate, as attacks could alter perceived quality. We evaluate the method from (Chistyakova et al., 2024) with different parameters and NT method (Liu et al., 2024) that employ gradient normalization during training. Both these methods are specifically designed for the IQA task.

*Table 1.* Evaluated defense methods in our benchmark by types (purification, adversarial training, certified methods).

| | Defense method | Short description | | Defense method | Short description |
|---|---|---|---|---|---|
| Adversarial purification | Gaussian blur | Smooth with a Gaussian filter | Adversarial training | Classic adv. training (Chistyakova et al., 2024) | Model fine-tuning on adv. img. |
| | Median blur | Smooth with a median filter | | Gradient Norm optimization (Liu et al., 2024) | Perform gradient normalization during training |
| | JPEG (Guo et al., 2018) | JPEG compression algorithm | | | |
| | Color quantization (Xu et al., 2018) | Reduce the number of colors | | | |
| | DiffJPEG (Reich et al., 2024) | Differentiable JPEG | | | |
| | Unsharp masking | Unsharp mask | | | |
| | FCN (Gushchin et al., 2024) | Neural filter to counter color attack | | | |
| | Flip | Mirror the image | | | |
| | Bilinear Upscale | Resize and upscale to original size | | | |
| | Resize (Guo et al., 2018) | Change the image size | | | |
| | Random Rotate | Image rotation | Certified | Random. Smoothing (RS) (Cohen et al., 2019) | Noisy samp. →clf.→voting |
| | Random Crop (Guo et al., 2018) | Crop the image | | Denoised RS (DRS) (Salman et al., 2020) | Noisy samp.→denoiser→ clf.→voting |
| | Random noise | Add random noise | | Diffusion DRS (DDRS) (Carlini et al., 2022) | Noisy samp.→1-step diffus.→clf.→voting |
| | MPRNet (Zamir et al., 2021) | 3-stage CNN for denoising | | DensePure (DP) (Chen et al., 2022b) | Noisy samp.→N-step diffus.→clf.→voting |
| | Real-ESRGAN (Wang et al., 2021) | GAN-based super-res. denoising | | Median Smoothing (MS) (Chiang et al., 2020) | Noisy samp.→reg.→median |
| | DISCO (Ho & Vasconcelos, 2022) | Enc.+loc. implicit module denoising | | Denoised MS (DMS) (Chiang et al., 2020) | Noisy samp.→denoiser→reg.→ median |
| | DiffPure (Nie et al., 2022) | Diffusion denoising | | | |

**Certified defenses** provide theoretical guarantees for the attacked model. The most common certified defenses are based on randomized smoothing. They can be applied to any IQA metric without restricting the model architecture. Certified defense methods generate noisy variations of the input images, which then pass through the model. Before passing them through the model, some methods apply denoising to boost accuracy. Table 1 provides further details. For each certified defense method, we generated 1000 noisy images as input for the metric. Currently, most smoothing methods are developed for classification, and one study (Chiang et al., 2020) investigated smoothing for regression. To apply the classification-based smoothing for IQA metrics, we are converting the IQA metric into a multiclass classification model with ordered classes (Hammoudeh & Lowd, 2023). Despite the challenge of IQA metrics discretization, classifier-based smoothing methods can yield impressive results for our task since they are more extensively studied. For a classification-based certified method, the output is a quality class and certified radius $R$; for a regression-based method, it's a metric score and certified delta. For more details refer to Sections 3.5 and A.3.5.

### 3.4. Experimental setup

**Datasets**. To thoroughly evaluate adversarial defenses we use four datasets. KonIQ-10k (10,073 images) (Hosu et al., 2020) and KADID-10k (10,125 images) (Lin et al., 2019) contain various natural images with multiple distortions. NIPS 2017: Adversarial Learning Development Set (2017, Competition Page, 1,000 images) is designed for evaluating adversarial attacks against image classifiers. AGIQA-3K dataset (Li et al., 2024) contains 2,982 AI-generated images for different quality levels. We randomly sampled 1,000 images from KonIQ-10k and KADID-10k datasets to balance computational efficiency and dataset diversity. We included each distortion type and strength and sampled 8 out of 81 original images from KADID, resulting in 1000 distorted images. Due to high computational complexity, we used 50 images from each dataset for black-box attacks and certified defenses. Appendix A.3.1 presents statistical tests verifying that our sampling procedure is valid and representative of the entire dataset.

**IQA metrics**. Based on the results of the IQA adversarial robustness benchmark (Antsiferova et al., 2024), we chose 9 NR IQA metrics: Meta-IQA (Zhu et al., 2020), MANIQA (Yang et al., 2022), CLIP-IQA+ (Wang et al., 2023), TOPIQ (Chen et al., 2024), Koncept (Hosu et al., 2020), SPAQ (Fang et al., 2020), PAQ2PIQ (Ying et al., 2020), Linearity (Li et al., 2020), and FPR (Chen et al., 2022a). These metrics employ different convolutional and transformer-based architectures and have a wide robustness $R_{score}$ range. The Appendix A.3.5 provides a more detailed description. Because of adversarial training's computational complexity, we selected only two NR IQA metrics, Linearity and Koncept, for their high correlation with subjective scores.

### 3.5. Evaluation metrics

**Robustness scores**. $R_{score}$ (Zhang et al., 2022a) and $R^{(D)}_{score}$ aim to assess model robustness by measuring relative score changes before and after attacks on a dataset of $N$ images. $R_{score}$ takes into consideration the maximum allowable quality-prediction change:

$$R_{score} = \frac{1}{N}\sum_{i=1}^{N}\log\left(\frac{\max\{\beta_1 - f_\omega(x_i), f_\omega(x_i) - \beta_2\}}{|f_\omega(x_i) - f_\omega(P(x_i'))|}\right), \tag{5}$$

where $x_i$ is the source image, $x_i'$ is the attacked version of $x_i$. $f_\omega(\cdot)$ is the IQA model, and $\beta_1$ and $\beta_2$ are the maximum and minimum of IQA scores in the dataset. In addition, we propose a variation of this metric called $R^{(D)}_{score}$, which differs only in applying purification $P(\cdot)$ to $x_i$ and $x_i'$. A larger value means better robustness.

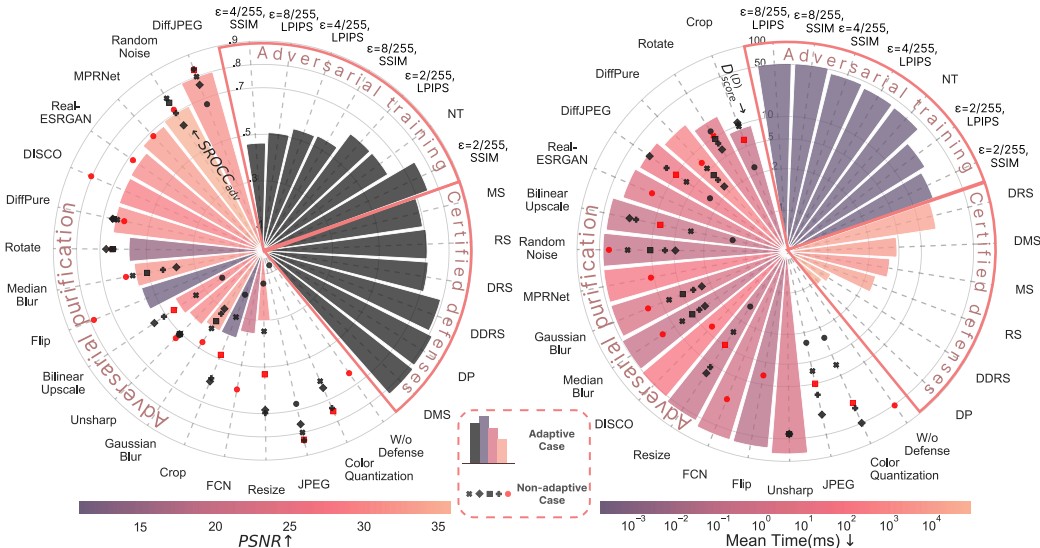

*Figure 2.* Adversarial defenses efficiency for IQA metrics in terms of $SROCC_{adv}$ (left) and $D_{score}^{(D)}$ (right). Bars and dots are for adaptive and non-adaptive attacks, respectively. Each dot represents the result for each preset of defense. Red dots represent a selected preset for the adaptive case. Results are averaged across 9 IQA metrics and 14 attacks.

We propose $D_{score}$ and $D_{score}^{(D)}$ to measure adversarial purification's ability to reduce the discrepancy between the IQA scores of the original and purified images:

$$D_{score} = \frac{100}{n} \sum_{i=1}^{n} \frac{|f_\omega(P(x_i')) - f_\omega(x_i)|}{\text{diam}(f_\omega)}; \quad (6)$$

$$D_{score}^{(D)} = \frac{100}{n} \sum_{i=1}^{n} \frac{|f_\omega(P(x_i')) - f_\omega(P(x_i))|}{\text{diam}(f_\omega)}, \quad (7)$$

where scores denoted with the superscript $^D$ are for purified source images, $P$ represents the purification method. Lower scores indicate better defense effectiveness, as they reflect smaller disparities between the IQA metrics of defended and original images. The metrics quantify how well a defense can restore the IQA scores of adversarial images to match their original values.

For certified defense methods, we additionally measured the *certified radius* ($Cert.R \uparrow$), which indicates how much the input image can undergo alteration without changing the class prediction; the *percentage of abstentions* ($Abst. \downarrow$), reported by classification-based methods when their predictions are highly uncertain; and *certified relative delta* ($Cert.RD \uparrow$), which is the certified delta, produced by the defense method, divided by diam($f_\omega$). This parameter characterizes how much a metric score can change in a fixed $l_2$ ball of norm $\epsilon$ around a given image $x$.

**Quality scores**. We use $PSNR$, $SSIM$(Wang et al., 2004), $MSE$ and $L_\infty$ to measure the perceptual similarity between purified images and their original images, reflecting the preservation of visual quality post-defense. The underlying principle is that the defense mechanism should restore the IQA score and preserve the image's perceptual quality. More complex IQA metrics (such as LPIPS) cannot be used in this environment due to possible transferabilities of adversarial attacks. We also conducted a subjective survey to better evaluate the quality of defended images.

**Performance scores**. We use $SROCC$ with MOS values $\vec{y}$ to assess an IQA metric's performance in the presence of adversarial defense. $PLCC$ results are similar to $SROCC$ and are provided in the Appendix A.6.

$$SROCC_{clear} = SROCC(\vec{y}, f_\omega(P(\vec{x}))); \quad (8)$$

$$SROCC_{adv} = SROCC(\vec{y}, f_\omega(P(\vec{x}'))) \quad (9)$$

### 3.6. Implementation details

We used a sophisticated end-to-end automated training and evaluation pipeline using GitLab CI/CD tools to ensure all our results are reproducible. All calculations required approximately 25,000 GPU-hours. Timing benchmarks were performed on a dedicated server with NVIDIA Tesla A100 80 Gb GPU, Intel Xeon Processor (Ice Lake) 32-Core Processor @ 2.60 GHz.

When available, we used original open-source implementations for all adversarial attacks, defenses, and IQA metrics. For each attack and defense, we varied one main parameter — commonly associated with the attack strength — while keeping the remaining parameters consistent with their original implementations (see Table 1). The Appendix A.3.3, A.3.5 provides a list of parameters for attacks and defenses and links to the original repositories.

*Table 2.* **Comparison of purification defenses**. Results are averaged for all images, attacks, IQA models for non-adaptive/adaptive cases.

| | Time (ms) ↓ | $SROCC_{clear}$ ↑ | $SROCC_{adv}$ ↑ | $D_{score}^{(D)}$ ↓ | $R_{score}$ ↑ | $PSNR_{adv}$ ↑ | $SSIM_{adv}$ ↑ | $MSE$ ↓, ×10⁻³ | $L_{inf}$ ↓ |
|---|---|---|---|---|---|---|---|---|---|
| W/o Defense | — | 0.511/0.511 | 0.413/0.413 | 66.68/66.68 | 0.56/0.56 | **44.61/44.61** | **0.94/0.94** | **2.51**/2.51 | **0.09/0.09** |
| Flip | **0.05** | 0.593/0.587 | 0.555/0.420 | 7.91/67.41 | **1.17**/0.45 | 10.76/10.76 | 0.28/0.29 | 110.47/109.80 | 0.95/0.95 |
| Color Quantization | 0.07 | 0.587/— | 0.532/— | 27.38/— | 0.83/— | 32.54/— | 0.86/— | 2.84/— | 0.11/— |
| Median Blur | 0.11 | 0.551/0.531 | 0.431/0.424 | 15.14/49.95 | 0.92/0.50 | 31.38/31.80 | 0.86/0.87 | 4.48/3.17 | 0.51/0.51 |
| Bilinear Upscale | 0.15 | 0.569/0.479 | 0.452/0.355 | 18.13/40.93 | 0.86/0.58 | 32.82/28.68 | 0.91/0.83 | 3.50/4.23 | 0.35/0.48 |
| Crop | 0.16 | 0.587/0.431 | 0.508/0.385 | 11.68/**6.49** | 0.92/0.78 | 11.53/11.00 | 0.33/0.37 | 89.94/105.34 | 0.94/0.93 |
| Resize | 0.19 | 0.597/0.511 | 0.549/0.353 | 10.56/54.31 | 1.02/0.42 | 32.11/29.38 | 0.90/0.85 | 3.83/3.90 | 0.37/0.45 |
| FCN | 0.52 | 0.571/0.562 | 0.478/0.310 | 23.89/64.32 | 0.80/0.41 | 20.89/20.78 | 0.78/0.77 | 13.24/13.35 | 0.54/0.55 |
| Unsharp | 0.78 | 0.611/0.595 | 0.427/0.370 | 43.22/80.24 | 0.52/0.32 | 30.34/29.77 | 0.87/0.86 | 3.81/3.03 | 0.33/0.35 |
| Gaussian Blur | 0.99 | 0.552/0.522 | 0.423/0.376 | 15.75/45.67 | 0.84/0.53 | 32.22/32.30 | 0.90/0.90 | 3.83/2.72 | 0.34/0.35 |
| Rotate | 2.14 | 0.560/0.585 | 0.501/0.469 | 6.64/16.24 | 1.09/0.89 | 11.56/14.65 | 0.31/0.42 | 96.44/54.03 | 0.97/0.96 |
| Real-ESRGAN | 5.89 | 0.552/0.501 | 0.503/0.436 | 9.47/30.13 | 0.66/0.58 | 30.32/30.47 | 0.89/0.88 | 3.97/2.98 | 0.43/0.44 |
| DiffJPEG | 8.11 | **0.625/0.610** | **0.608/0.549** | 12.94/29.81 | 1.07/0.71 | 34.33/31.33 | 0.91/0.87 | 3.04/2.61 | 0.26/0.33 |
| Random Noise | 8.29 | 0.556/0.594 | 0.539/0.508 | 10.14/44.84 | 0.87/0.59 | 25.42/35.87 | 0.54/0.90 | 4.78/**1.79** | 0.30/0.13 |
| MPRNet | 65.79 | 0.565/0.565 | 0.535/0.488 | 12.14/45.00 | 0.97/0.53 | 32.21/32.32 | 0.88/0.89 | 4.23/2.91 | 0.37/0.36 |
| DISCO | 139.60 | 0.585/0.562 | 0.581/0.476 | 3.51/47.91 | 1.14/0.50 | 29.12/29.08 | 0.86/0.86 | 4.34/3.31 | 0.43/0.43 |
| JPEG | 227.34 | 0.622/— | 0.605/— | 13.07/— | 1.07/— | 34.25/— | 0.90/— | 3.03/— | 0.26/— |
| DiffPure | 691.42 | 0.496/0.487 | 0.485/0.470 | **2.01**/22.96 | 0.79/0.75 | 27.59/30.11 | 0.79/0.86 | 5.34/3.44 | 0.48/0.43 |

## 4. Results

In all tables and figures, for non-adaptive cases, we report the results of defenses with a hyperparameter set that provides the best $SROCC_{adv}$. Table 2 shows overall results for adversarial purification defenses, and Table 3 — for adversarial training and certified methods.

Adversarial perturbations generally consist of high-frequency noise, making compression-based defenses particularly effective. DiffJPEG leads in terms of several evaluation metrics among purifications with the best $SROCC_{adv}$, $D_{score}$, and $R_{score}$. JPEG and DiffJPEG remove high-frequency noise alongside adversarial perturbations while preserving the structural information of a clean image, as the perturbation has a far more complex and unnatural representation than the natural high-frequency components. Thus, the developers of purification methods should analyze how the high-frequency components of an image and perturbations differ. DISCO, which uses an encoder-decoder architecture similar to compression, leverages learned features of clean images to project images back onto the natural manifold. Denoising methods, such as MPRNet and Real-ESRGAN, show average performance, as they were trained on noise of a more uncomplicated nature. At the same time, adversarial perturbations possess more complex high-frequency structures. Fine-tuning these methods on adversarial perturbations is a promising direction for future research. Diffusion-based models offer high variability in strength, allowing precise tuning for specific adversarial attack budgets. On the other hand, DiffPure introduces its own processing artifacts, causing the worst correlations and lower image quality of defended images. This highlights a key difference between applying diffusion-based defenses in classification tasks (where they are state-of-the-art methods) and IQA tasks, underscoring the need for task-specific adaptations.

Compared to purification methods, adversarial training provides superior correlations but shows worse $D_{score}$ and doesn't purify images. Certified methods deliver the best overall combination of correlations, $D_{score}$ and $R_{score}$, but are highly impractical due to computational overhead.

**Parameter Variations for Defenses and Attacks**. Figure 3 (a) illustrates how the parameters of adversarial purification methods impact the trade-off between robustness and performance in non-adaptive scenario. Strong defenses, located in the lower-left corner of the scatter plot, nearly restore IQA metric scores to their pre-attack values but significantly reduce correlations with subjective quality, making them impractical for real-world applications. The red line highlights the Pareto-optimal front, which includes strong JPEG compression, weak DiffPure, Gaussian blur, and DISCO.

The comparison results for different attack parameters, as presented in Appendix A.6 (Table 11), show that increasing attack strength leads to decreased defense success. Notably, defenses in non-adaptive case remain stable across different attack strengths, while in the adaptive case they are highly sensitive to attack intensity. Strong attacks cause significant correlation decrease, making most defenses impractical in real-world applications, with DiffPure and DiffJPEG being notable exceptions. Importantly, the relative ranking of defenses remains mostly consistent across attack strengths.

**Inference computational complexity**. Tables 2 and 3 present time costs across defenses. Certified methods excel in defense efficiency but are computationally intensive, with the fastest certified method being 4× slower than the slowest purification one. Adversarial training requires no inference overhead. For purification-based defenses, computational demands vary. Basic preprocessing (blurring and rotation) has minimal overhead, while diffusion-based methods (e.g., DiffPure), are much slower due to multiple denoising steps.

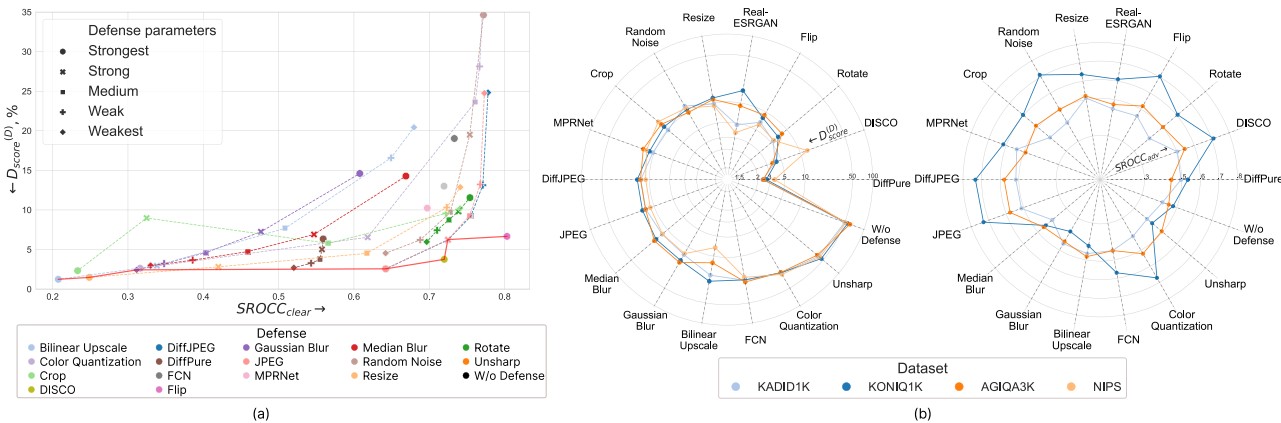

*Figure 3.* (a) Scatter plot of purification defense parameter configurations in the non-adaptive case, with the red line indicating Pareto-optimal defenses. (b) Performance across test datasets in terms of $D_{score}^{(D)}$ (left) and $SROCC_{adv}$ (right).

*Table 3.* Comparison of adversarial training (AT) (left) and certified defenses (right). **C**: classification-based methods, **R**: regression-based. For AT methods, APGD is an attack used for fine-tuning; 2/4/8 is perturbation budget. LPIPS/SSIM are FR metrics for MOS adjustment.

| | Adaptive attacks, 1000 images KonIQ-10k, Koncept+Linearity | | | | | Non-adaptive attacks, 10 images from KonIQ-10k, 9 IQA metrics | | | | | |
|---|---|---|---|---|---|---|---|---|---|---|---|
| AT Defense | $SROCC_{clear}\uparrow$ | $SROCC_{adv}\uparrow$ | $D_{score}^{(D)}\downarrow$ | $R_{score}\uparrow$ | Cert Defense | Time(ms)↓ | $SROCC_{clear}\uparrow$ | $SROCC_{adv}\uparrow$ | $D_{score}^{(D)}\downarrow$ | $R_{score}\uparrow$ | $Cert.R\uparrow / Cert.RD\downarrow$ |
| APGD-LPIPS-2 | 0.840 | 0.651 | 20.70 | 1.10 | RS (**C**) | 11080 | 0.747 | 0.706 | 2.70 | 5.61 | **0.183** / $\infty$ |
| APGD-LPIPS-4 | 0.866 | 0.576 | 36.59 | 0.77 | DRS (**C**) | 15320 | **0.882** | 0.712 | 16.57 | 2.01 | 0.175 / $\infty$ |
| APGD-LPIPS-8 | 0.867 | 0.547 | 45.61 | 0.69 | DDRS (**C**) | 39800 | 0.819 | 0.792 | 1.20 | **6.21** | 0.174 / $\infty$ |
| APGD-SSIM-2 | 0.830 | **0.763** | **17.11** | **1.23** | DP (**C**) | 82130 | 0.823 | 0.815 | **1.09** | 6.20 | 0.162 / $\infty$ |
| APGD-SSIM-4 | 0.852 | 0.625 | 39.53 | 0.80 | MS (**R**) | **2830** | 0.753 | 0.694 | 3.80 | 1.92 | 0 / 1.707 |
| APGD-SSIM-8 | **0.873** | 0.582 | 45.38 | 0.64 | DMS (**R**) | 5970 | 0.875 | **0.822** | 4.70 | 1.89 | 0 / **1.440** |
| NT (Liu et al., 2024) | 0.815 | 0.649 | 35.42 | 0.81 | | | | | | | |

**Defenses against adaptive and non-adaptive attacks**. Figure 2 and Table 2 compare defense performance against non-adaptive and adaptive attacks. Adversarial training was measured only in adaptive setting, while certified defenses were only for non-adaptive scenario. By design, adaptive attacks are significantly more successful, so $D_{score}^{(D)}$ robustness bars are higher than markers on Figure 2. Furthermore, the $SROCC_{adv}$ of defended IQA metrics is lower, due to the more unpredictable behavior of adaptive attacks. Simple spatial transformations (Flip, Resize) and frequency filtering (Gaussian Blur, Median Blur) are effective in the non-adaptive case but insufficient for adaptive one. In adaptive case, Crop, Rotate and DiffPure excel in $D_{score}^{(D)}$ and $R_{score}$, suggesting high randomness is crucial for effective defense. The combination of randomness and geometric transformations particularly mitigates perturbations. Flip and Random Rotate are great examples: the first lacks randomness, and adaptive attacks easily surpass it, while Random Rotate reduces attack effectiveness since the angle differs between attack calculation and inference. Specialized defenses can demonstrate high effectiveness in the non-adaptive case but exhibit unpredictable performance against adaptive attacks. For instance, DISCO ranks top-3 by $R_{score}$ in the non-adaptive case but plummets under adaptive attacks, whereas DiffPure maintains its top position in both scenarios.

**Defenses for regular/AI-gen image content** Figure 3 (b) compares $D_{score}$ and $SROCC_{adv}$ on different datasets, including three natural-scene image datasets and the AI-generated images from AGIQA-3K. There is no significant difference in defense efficiency for most methods between datasets, but some advanced defenses based on neural networks (Real-ESRGAN, DISCO) have larger discrepancies. This highlights the critical importance of dataset coverage of the target data domain during training. As shown in Figure 3 (b, right), correlations depend highly on the dataset. On average, $SROCC_{adv}$ on KonIQ-1k is significantly higher than on KADID and AGIQA-3K, similar to results in Table 19 in the Appendix regarding $SROCC_{clear}$. This can be due to two factors: a) Several IQA models (e.g., TOPIQ and CLIP-IQA+) were trained on the KonIQ-10k dataset or its subsets, giving them a natural advantage. b) Certain IQA models, such as MetaIQA and PAQ2PIQ, generally achieve higher correlation values on KonIQ-10k, as reported in their respective studies, suggesting an inherent dataset bias.

**Guarantees of the defenses.** Among all the methods compared, only certified methods provide theoretically reliable predictions. Table 3 presents the results for the certified defenses. Compared to more sophisticated methods, simple randomized smoothing showed the highest certified radius. Among regression-based methods, Denoised Median

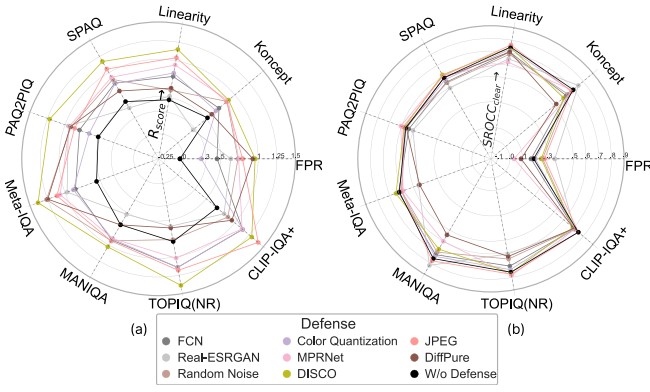

Figure 4. $R\,score$ (a) and $SROCC_{clear}$ (b) on different IQA metrics of some purification defenses.

Smoothing showed the lowest certified relative delta. Although there are no theoretical restrictions on using certified defenses without denoising, our experiments indicate that the denoising step is crucial to obtain a defended model that produces high SROCC for the IQA task. Another finding is that, despite the questionable applicability of randomized smoothing-based defenses to IQA, they remain effective, as the certified radii are sufficiently high and the number of abstentions is relatively low.

**Defenses for different IQA metrics' architectures**. The chosen IQA metrics fall into categories by their backbones: CNN-based (Meta-IQA, Koncept, SPAQ, PAQ2PIQ, Linearity), transformer-based (MANIQA, CLIP-IQA+, TOPIQ), and custom (FPR). Figure 4 shows $R_{score}$ and $SROCC_{clear}$ for these metrics in a non-adaptive scenario.

Transformer-based metrics have greater $R_{score}$ robustness even without defense. Their self-attention mechanisms enable comprehensive global contextual analysis, capturing image-wide dependencies beyond local features. This architectural feature provides an intrinsic resilience to subtle adversary perturbations, resulting in a lower robustness increase when defenses are applied. For other architectures, most defenses increased the robustness. DISCO improved the robustness of all metrics, but the effect was much stronger on CNN-based metrics. Defended transformer-based metrics showed a higher correlation decrease than metrics of other architectures. Note that custom architectures can be highly vulnerable (FPR model shows the worst $R_{score}$). This vulnerability is likely caused by its atypical architecture with a Siamese network and an attempt to "hallucinate" the features of the pseudo-reference image from a distorted one. These results correlate with previous research (Antsiferova et al., 2024). In general, all tested defenses do not impose restrictions on the architecture of the models; however, some defenses perform better for specific metrics.

**Perceptual quality of defended images**. Most presented purification defenses aim to restore the original content of the image, but inevitably introduce artifacts. The most noticeable ones include loss of details (DISCO, MPR-Net), altering the image content (Real-ESRGAN, Diff-Pure), blurring (DiffPure, blur), and compression artifacts (JPEG/DiffJPEG, Color Quantization). Figures 9 and 8 in the Appendix A.5 show examples of images with such artifacts. Table 2 shows quantitative results of perceptual quality. Attacked images turned out to be closer to clean images than purified ones. $PSNR$ and $SSIM$ cannot account for geometric transformations and, thus, are meaningless for Flip, Rotate, and Crop.

To assess perceptual quality of defended images we conducted a large-scale crowd-source subjective study with 2,700+ participants who provided 60,000+ responses (details in Appendix A.2). The results highlight Real-ESRGAN as the top method, achieving perceptual quality that surpasses even the "W/o defense" case in quality. This method treats adversarial perturbations as degradations, effectively removing them while preserving natural image content. Real-ESRGAN has architecture and training objective that naturally suppress high-frequency noise, a common feature of adversarial attacks, and demonstrate a unique ability to mitigate unrestricted attack perturbations — an area where traditional denoising methods falter. The model's relatively lower performance on objective metrics can be attributed to aliasing artifacts introduced during downsampling operations, residual noise, and subtle change in contrast. Subjective evaluation reveals that Gaussian blur and DiffJPEG perform significantly worse than their objective metrics suggest. While Gaussian blur fails to suppress both high-frequency noise and unrestricted attack artifacts, DiffJPEG introduces strong blocking artifacts and texture loss. DiffPure not only fails to suppress high-frequency noise, but also distorts textures and blurs image, resulting in low subjective scores. These findings highlight the importance of subjective evaluations: objective metrics such as SSIM and PSNR can misrepresent perceived quality and thus lead to misleading leaderboard rankings.

**Statistical tests**: A one-sided Wilcoxon signed-rank test with Bonferroni correction (see Appendix A.4) further proves the statistical significance of the results obtained in our comparison and confirm that DISCO, DiffPure and DiffJPEG defenses significantly outperform all others in both $D_{score}^{(D)}$ and $R_{score}$ with consistently high pairwise win-rate percentages across datasets. In the adaptive setting, APGD-SSIM-2 adversarial training also demonstrates a statistically significant robustness advantage over competing defenses.

# 5. Conclusion

This paper introduces the first comprehensive benchmark for evaluating defenses against adversarial attacks on neural network-based Image Quality Assessment (IQA) metrics, addressing a critical gap in the field. By systematically analyzing 30 defense strategies across 14 attack methods and 9 IQA metrics, we provide a framework to guide the development of secure and reliable IQA models.

Our results highlight superiority of compression-based defenses due to their ability to remove high-frequency adversarial perturbations while preserving the structural details. This suggests that future defenses should focus on better distinguishing genuine high-frequency details from adversarial noise. Diffusion-based defenses, such as DiffPure, demonstrate strong performance in classification tasks but struggle in IQA tasks, emphasizing the need for task-specific adaptations of defenses and the importance of diverse and representative training datasets. Furthermore, the results highlight the critical role of randomness in mitigating attacks in the adaptive case.

We conducted a large-scale subjective study with 60,000+ responses to evaluate the perceptual quality of purification defenses and collect Mean Opinion Score values for a new dataset, which serves as a valuable resource for adversarial training and detection of adversarial perturbations.

Robust IQA metrics are essential for applications such as search engine optimization, video processing, and benchmarking, where adversarial vulnerabilities threaten trust and fairness. By publishing a dataset, detailed results, and an online leaderboard, we establish transparent and practical foundation to the research community and industry, enabling the development of more secure IQA models.

# Impact Statement

This paper presents work whose goal is to advance the field of Machine Learning. There are potential societal consequences, none which we feel must be highlighted here.

ACKNOWLEDGMENTS

This work was supported by a grant, provided by the Ministry of Economic Development of the Russian Federation in accordance with the subsidy agreement (agreement identifier 000000C313925P4G0002) and the agreement with the Ivannikov Institute for System Programming of the Russian Academy of Sciences dated June 20, 2025 No. 139-15-2025-011.

The research was carried out using the MSU-270 supercomputer of Lomonosov Moscow State University. We also would like to express our gratitude to Mikhail Pautov for discussing the results of this research.

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

# A. Appendix

## A.1. Limitations

While the proposed framework for benchmarking defenses against adversarial attacks on IQA metrics offers significant contributions, we acknowledge the existence of the following limitations to be addressed in future work:

1. **Handling Multiple Parameter Attacks**: The current framework deals mainly with attacks that have a single parameter. However, Some attacks might have multiple parameters to control their strength, complicating the evaluation process. Moreover, a group of Boundary Attacks adapts their parameters according to the response of the attacked model, which poses an additional challenge in a fixed-parameter setting. Future versions will include methods for dealing with different types of evolving attacks, possibly through dynamic parameter optimization techniques

2. **Transferability of Adversarial Attacks**: There might be defenses that better generalize to attacks produced on other defenses. Currently, the framework does not evaluate the transferability of adversarial attacks among different defenses. Future versions should provide insights into the generalizability and robustness of the defense.

3. **Simplified Ranking Methodology**: The current framework employs a straightforward ranking methodology that may not fully capture the complexity and the existence of different evaluation metrics with varying importance levels depending on the attack used for testing. Different evaluation measures can be assigned different weights based on their importance and relevance to the attack. This system allows for a composite score that reflects the overall performance of a defense mechanism. To provide a nuanced assessment of metric robustness, a more rigorous statistical framework for ranking metrics will be employed in future versions.

Addressing these limitations in future work will ensure the framework's robustness and adaptability in diverse and realistic scenarios.

## A.2. Subjective study

To assess the perceptual quality of defended images we have conducted a large-scale crowd-sourced subjective study on the Subjectify.us, a platform that employs the Bradley-Terry model to convert pairwise comparisons into numerical scores. We selected 5 attack methods in adaptive setting (Zhang et. al.-DISTS, IFGSM, Korhonen et. al., UAP), 4 IQA models (Linearity, MANIQA, TOPIQ, CLIP-IQA+), and 12 purification defenses. Non-differentiable methods (Color quantization, JPEG) and defenses with obvious perceptual differences (FCN, Crop, Flip, Rotate) were excluded from the study. The attacks were applied in an adaptive setting to 20 randomly selected source images, resulting in 5,200 images to compare (4 IQA model * 5 attacks * (12 defenses + 1 W/o defense) * 20 source images = 5200).

Each participant was given this instruction: "You will be shown one original image and two modified ones. For each pair of selected images, select the one that seems to you to be of higher quality, more realistic, and closer to the original image".

Because the number of pairwise comparisons grows quadratically with the number of images to compare, we divided the dataset into 400 subjective evaluations, each focusing on a single source image, attack, and IQA model combination. Each pair received at least 10 votes, with responses from participants failing verification questions being discarded. Bradley-Terry scores were computed for each defense method per evaluation and averaged across 400 subjective evaluations to determine the final rankings.

Figure 5 illustrates the results of this experiment. Subjective scores are along X-axis, $R_{score}$ are across Y-axis.

## A.3. Details of methodology

### A.3.1. DATASETS

In Table 4 we provide information about the datasets used in our study.

We employ 4 datasets: KonIQ-10k (10,073 images) (Hosu et al., 2020) and KADID (10,125 images) (Lin et al., 2019), NIPS 2017: Adversarial Learning Development Set (1000 images) (2017, (Competition Page)), AGIQA-3K (2982 images) (Li et al., 2024). KonIQ-10k, KADID and AGIQA-3K contain MOS scores that is used in evaluating correlations with ground-truth labels. The datasets were chosen to represent different distortion and generation types.

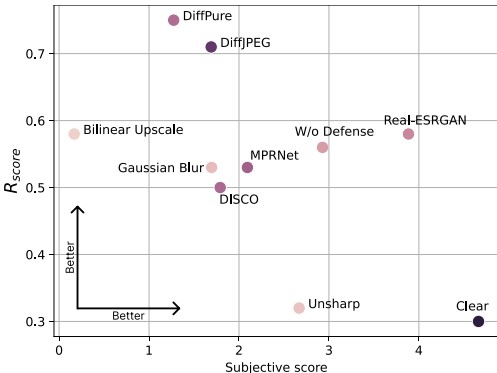

*Figure 5.* Results of subjective evaluation. Subjective scores are along X-axis, $R_{score}$ are across Y-axis.

*Table 4.* List of datasets used in our benchmark. These dataset consist of clean images. Based on them, we constructed a new dataset of adversarial images with MOSes, that consists of 5,200 images.

| Dataset | Size | Resolution | Subjective ratings | Short description |
|---|---|---|---|---|
| KonIQ-10k | 1,000 (out of 10,073) | $512 \times 384$ | 120,000 | Provides wide range of real-world photos with authentic distortions |
| KADID-10k | 8 out of 81 original images | $512 \times 384$ | 30,000 | Large-scale dataset with wide variety of content and artificial distortions |
| NIPS 2017 | 1,000 | $299 \times 299$ | — | Competition on adversarial examples and defenses in the NIPS 2017 |
| AGIQA-3K | 2,982 | $512 \times 512$ | 125,244 | AGIs from GAN-/auto-regression-/diffusion-based model with subjective scores |

We use subsets of KonIQ-10K (Hosu et al., 2020) and KADID (Lin et al., 2019) to evaluate adversarial defense methods. Both subsets of KonIQ-10K and KADID contain 1000 images at $512 \times 384$ resolution. KADID subset contains all distortions from 8 images, resulting in 1000 total images, that contains all distortions and do not decrease its diversity. Original KonIQ-10K dataset was partitioned into 10 clusters using K-Means (Lloyd, 1982) based on 3 parameters: Spatial Information (SI), Colorfulness (CF), and Mean Opinion Scores (MOS). We selected 100 random images from each cluster, resulting in a diverse set of 1000 test images regarding quality and content. Due to significantly higher computational complexity, we used a smaller set of 50 images for black-box attacks. They were sampled 5 images from each out of 10 clusters using K-Means (Lloyd, 1982) based on 3 parameters: Spatial Information (SI), Colorfulness (CF), and Mean Opinion Scores (MOS) where possible. For the same reason, we used a smaller set of 10 images for certified defenses to generate attacks. To evaluate the impact of sampling this procedure was repeated 10 times, focusing on purification methods and black-box attacks to accelerate calculations. We used default parameters for defense methods and 3 presets from the main paper for attacks. For each IQA model, attack and defense method, we calculated four scores per sample: $D_{score}$, $SROCC_{clear}$, $SROCC_{adv}$, and $SSIM$.

Figure 6 illustrates the distribution of these scores for each sample for KonIQ-10k dataset. The results show that the distributions are nearly identical across all samples and metrics, with consistent mean values. To assess the differences between the means of distributions, we computed the mean for each distribution and score, yielding a list of 10 mean values per score. Then, we calculated the mean and variance of these values across the 10 samples. These values can be found in Table 5. To verify these findings statistically, we performed a Kruskal-Wallis test (Kruskal & Wallis, 1952) for each metric across the 10 samples. The p-values are shown in Table 5. These p-values indicate no significant differences between the samples, confirming that the sampling procedure does not introduce variability into the evaluation results. This consistency strengthens our conclusions and ensures that the findings are robust across different random subsets of the dataset.

A.3.2. IQA METRICS

We define metric range as $\mathrm{diam}(f_\omega) = \sup\limits_{x,z \in X} \{|f_\omega(x) - f_\omega(z)|\} = \text{upper} - \text{lower}$, where upper is called the upper metric bound and lower - lower metric bound. To calculate these bounds, we used the DIV2K_valid_HR subset from the DIV2K dataset (Agustsson & Timofte, 2017). The upper bound is set to the highest metric value across the chosen subset, while the lower bound is set to the minimum value between the lowest metric value on subset images compressed with JPEG with quality of 10 and sampled random noise of the image subset size.

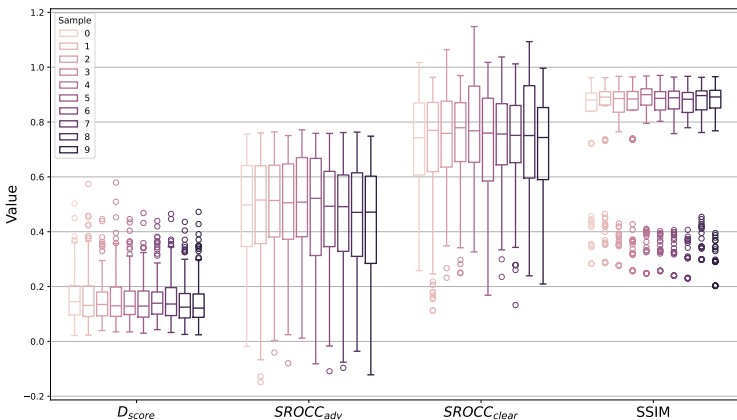

*Figure 6.* The effect of sampling 50 images on results. Each box represents one experiment.

*Table 5.* **Validity of sampling methodology**. To validate our sampling strategy we compare results of defenses on 10 different samplings of 50 images. Results of the Kruskal-Wallis test suggests that samplings have no significant differences.

| Score | Mean | Variance of means for each sample | p-value after Kruskal-Wallis test |
|---|---|---|---|
| $D_{score}$ | 0.1533 | 0.000044 | 0.5425 |
| $SROCC_{adv}$ | 0.4681 | 0.00043 | 0.1449 |
| $SROCC_{clear}$ | 0.7343 | 0.00066 | 0.1958 |
| $SSIM$ | 0.7953 | 0.000034 | 0.1138 |

*Table 6.* List of NR IQA metrics used in our benchmark.

| Metric | $R_{score} \uparrow$ | Backbone | Number of parameters | Input transformations | Bounds | Code |
|---|---|---|---|---|---|---|
| Meta-IQA (Zhu et al., 2020) | 1.168 | ResNet-18 | 13.2M | ImageNet Normalization | 0.00/1.00 | Github |
| MANIQA (Yang et al., 2022) | 0.986 | ViT-B/8 | 135.62M | $224 \times 224$ crop | 0.00/1.00 | Github |
| Koncept (Hosu et al., 2020) | 0.584 | InceptionResNetV2 | 59.82M | Normalization (0.5, 0.5) | 26.40/66.87 | Github |
| SPAQ (Fang et al., 2020) | 0.493 | ResNet-50 | 23.5M | $224 \times 224$ crop | 21.75/77.75 | Github |
| PAQ2PIQ (Ying et al., 2020) | 0.449 | ResNet-18 | 11M | — | 58.38/84.17 | Github |
| Linearity (Li et al., 2020) | 0.267 | ResNeXt-101 | 90M | ImageNet Normalization | 25.78/83.23 | Github |
| FPR (Chen et al., 2022a) | -0.229 | Custom | 16.6M | Splitting into fixed-sized patches | 47.22/77.05 | Github |
| CLIP-IQA+ (Wang et al., 2023) | 0.713 | CLIP | 244M | — | 0.00/1.00 | Github |
| TOPIQ (Chen et al., 2024) | 0.865 | Transformer | 45M | — | 0.22/0.82 | Github |

We report metric ranges and parameters in Table 6. The $R_{score}$ is taken from (Antsiferova et al., 2024).

### A.3.3. USED ADVERSARIAL ATTACKS

Early methods for attacking IQA metrics aimed to stress-test performance. (Wang & Simoncelli, 2008) introduced the **MADC** method, which uses gradient projection onto a proxy FR metric to compare accuracy. Later, (Kurakin et al., 2018) proposed **I-FGSM**, iteratively adding gradients to the image, but this caused visible distortions. (Korhonen & You, 2022) addressed this by targeting high-textured regions using Sobel-filter-based weighting (**Korhonen et al.**), while (Zhang et al., 2022b) incorporated FR IQA metrics like DISTS and LPIPS into loss functions (**Zhang et al.**). **SSAH** by (Luo et al., 2022) limited attacks to high frequencies, and (Bhattad et al., 2019) introduced **cAdv**, operating in LAB color space. Efficient universal perturbation methods, such as **UAP** and **FACPA**, eliminate backpropagation during inference, with FACPA further optimizing for high-resolution data.

*Table 7.* List of adversarial attacks used in our benchmark. WB and BB are white-box and black-box attack types. We adjust varied parameters to align attacks' strengths.

| Adversarial attack | Type | Restriction | Varied parameter | Short description |
|---|---|---|---|---|
| I-FGSM (Kurakin et al., 2018) | WB | $l_\infty$ | lr | Grad. descent to increase IQA metric |
| Optimised-UAP (Shumitskaya et al., 2024) | WB | $l_\infty$ | amplitude | Universal perturb. via grad. descent |
| Korhonen et al. (Korhonen & You, 2022) | WB | $l_\infty$ | lr | Sobel-filter-masked gradient descent |
| Zhang et al. (Zhang et al., 2022b) | WB | $l_\infty$ | lr | Grad. descent with saving DISTS |
| MADC (Wang & Simoncelli, 2008) | WB | $l_\infty$ | lr | Grad. project. onto MSE |
| cAdv (Bhattad et al., 2019) | WB | SSIM | lr | Grad. descent with recolorization |
| SSAH (Luo et al., 2022) | WB | $l_\infty$ | lr | Grad. descent with high-freq. min. |
| FACPA (Shumitskaya et al., 2023) | WB | $l_\infty$ | amplitude | Perturb. generated using U-Net |
| NES (Ilyas et al., 2018) | BB | $l_\infty$ | $\epsilon$ | Grad. descent with approx. gradient |
| Parsimonious (Moon et al., 2019) | BB | PSNR | $\epsilon$ | Perturbs using discrete optimization |
| One Pixel (Su et al., 2019) | BB | $l_0$ | pixel count | Perturbs pixels with diff. evolution |
| Square (Andriushchenko et al., 2020) | BB | $l_\infty$ | $\epsilon$ | Square-like perturb. via rand. search |
| Patch-RS (Croce et al., 2022) | BB | PSNR | $\epsilon$ | Finds adv. patch via random search |

For black-box attacks, we adopted efficient methods originally designed for image classifiers. **NES** ((Ilyas et al., 2018)) estimates gradients using natural evolutionary strategies, while the **Parsimonious attack** ((Moon et al., 2019)) identifies sparse pixel-level perturbations through hierarchical updates. **Square attack** ((Andriushchenko et al., 2020)) applies square patches in a random search algorithm. Representing sparse attacks, **Patch-RS** ((Croce et al., 2022)) uses random search to place patches. Finally, **One Pixel** ((Su et al., 2019)) uses the Differential Evolution algorithm to alter minimal pixels.

Descriptions of these methods and varied parameters are detailed in Table 7.

### A.3.4. CHOOSING PARAMETERS FOR ATTACKS

To align attacks by strength across different metrics and defenses, we developed the methodology illustrated in Figure 7. For each attack, a restriction metric was selected: $PSNR$, $SSIM$ or $L_\infty$. While $L_\infty$ is suitable for most attacks, $SSIM$ is more appropriate for color-based attacks. For unrestricted attacks, such as Parsimonious and One Pixel, $PSNR$ is the most suitable. For these restriction metrics we defined three sets of target values corresponding to "weak", "medium" and "strong" attacks. Specifically:

- For $L_\infty$, the values $\frac{2}{255}$, $\frac{4}{255}$ and $\frac{8}{255}$ were chosen to be consistent with previous studies

- For $SSIM$, 0.75, 0.8 and 0.9 were chosen

- For $PSNR$, the values 25, 30 and 35 were used

For each attack, a single parameter corresponding to attack strength was varied (see Table 7). We computed the mean attack strength on the KonIQ-10k dataset for 10 values of this parameter and performed a linear approximation to match the target strength values.

### A.3.5. EVALUATED DEFENSES

**Purification**

According to (Guo et al., 2018), several standard image preprocessing techniques can be used as defenses against additive adversarial noise. These methods include compression (**JPEG**, **DiffJPEG** (Reich et al., 2024) **color quantization** (Xu et al., 2018)), spatial transformations (**Resize**, **Rotate**, **Crop**, **Flip**), blurring (**Median blur**, **Gaussian blur**, etc.), **Unsharp masking**, and others. Although not originally designed for adversarial defense, studies have demonstrated that these methods can be effective.

Since adversarial perturbations often consist of high-frequency noise, denoising techniques can be particularly useful. The Multi-Stage Progressive Image Restoration Network (**MPRNet** (Zamir et al., 2021)) is a three-stage convolutional neural

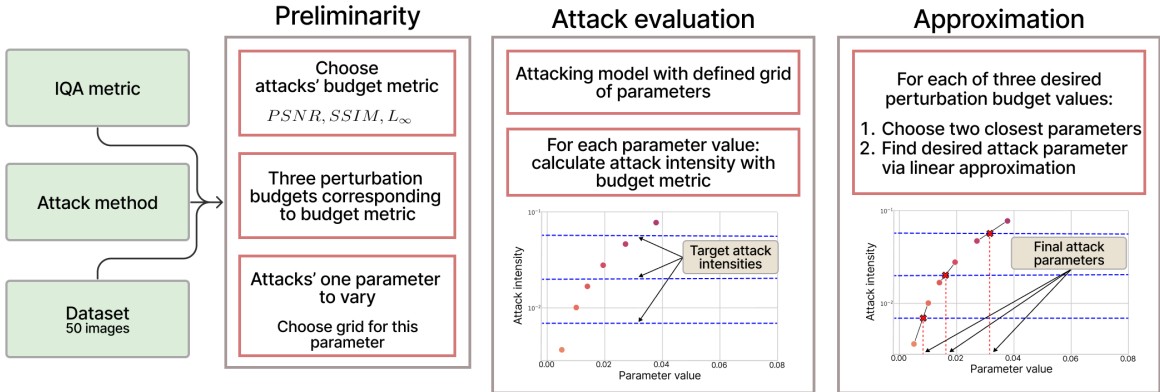

*Figure 7.* Procedure for selecting adversarial attack parameters.

*Table 8.* List of compared adversarial Purification methods.

| Defense method | Type | Varied parameter | Varied parameter values | Fixed parameters | Code |
|---|---|---|---|---|---|
| JPEG (Guo et al., 2018) | Compression | q | 10, 30, 50, 70, 90 | — | — |
| DiffJPEG (Reich et al., 2024) | Compression | q | 10, 30, 50, 70, 90 | — | Github |
| Color quantization (Xu et al., 2018) | Compression | npp | 2, 5, 16, 20, 25 | — | — |
| Resize (Guo et al., 2018) | Spat. transform. | scale | 0.1, 0.25, 0.5, 0.75, 0.9 | — | — |
| Bilinear Upscale | Spat. transform. | scale | 0.1, 0.25, 0.5, 0.75, 0.9 | — | — |
| Rotate | Spat. transform. | angle lim. | 10, 15, 20, 30, 50 | — | — |
| Crop (Guo et al., 2018) | Spat. transform. | size | 32, 64, 128, 256, 288 | — | — |
| Flip | Spat. transform. | — | — | — | — |
| Gaussian blur | Blurring | kernel size | 3, 5, 7, 9, 11 | sigma=0.15*kernel_size+ 0.35 | — |
| Median blur | Blurring | kernel size | 3, 5, 7, 9, 11 | — | — |
| Unsharp masking | Preprocessing | kernel size | 3, 5, 7, 9, 11 | sigma=1, amount=1 | — |
| MPRNet (Zamir et al., 2021) | Denoising | — | — | — | Github |
| Real-ESRGAN (Wang et al., 2021) | Denoising | — | — | denoise_strength=0.2, outscale=1, tile=0, tile_pad=10, pre_pad=0 | Github |
| DiffPure (Nie et al., 2022) | Defense | t | 5, 10, 20, 30, 50 | t_delta=15, diffusion_type=ddpm, sample_step=1 | Github |
| DISCO (Ho & Vasconcelos, 2022) | Defense | — | — | — | Github |
| FCN (Gushchin et al., 2024) | Defense | — | — | — | Github |
| Random noise | Adding noise | — | — | — | — |

network for image deblurring, deraining, and denoising. The first two stages use an encoder-decoder architecture for multi-scale contextual information, while the final stage operates at the original resolution to preserve details. MPRNet features supervised attention modules and cross-stage feature fusion for effective information transfer. **Real-ESRGAN** (Wang et al., 2021), a GAN-based model with several residual dense blocks for super-resolution, is trained with synthetic data and can be used for adversarial denoising.

We also included methods designed specifically as adversarial defenses for image classifiers or IQA models. **DiffPure** (Nie et al., 2022) employs diffusion models to purify adversarial images by introducing a small amount of noise through forward diffusion, and then reversing the process to recover a clean image. **DISCO** (Ho & Vasconcelos, 2022) is an image purification method aimed at enhancing classification robustness. It employs local implicit functions to ensure small perturbations do not significantly alter local data representations. By maintaining these robust local representations, DISCO effectively resists adversarial perturbations that do not align with the data's local structure. Some adversarial attacks, such as color-based modifications, are not bounded. Standard denoising approaches are ineffective against these. (Gushchin et al., 2024) proposed neural filter **FCN**, to counter color attack cAdv on image quality metrics. FCN features a compact, fully convolutional architecture with three hidden layers of 64, 32, and 3 filters

We report the list of parameters for defenses and fixed values for non-varied parameters in Table 8.

**Adversarial training** We applied the method proposed in (Chistyakova et al., 2024). More specifically, we fine-tuned Linearity and Koncept IQA models using the original images and attacked images in a 1:1 ratio from the original KonIQ-10K training dataset for 30 epochs. During the training process, we used a 2-step APGD attack (Croce & Hein, 2020) to generate

the attacked images. This method uses an adaptive step that allows a small number of iterations to achieve strong adversarial examples and reduce computational time. The goal of the attack during the training process is to increase model loss. We adjusted the MOS values based on the FR metric scores. For a given original image $x$ with MOS $y$, we obtain the adjusted MOS for the attacked image $x'$ as follows:

$$y' = y - M(x, x') \tag{10}$$

We have considered LPIPS and 1 - SSIM as M. To evaluate the impact of attack magnitude during training we chose 3 different attack magnitudes $\varepsilon = \{2, 4, 8\}/255$.

We also evaluated method proposed in (Liu et al., 2024). Specifically, the approach introduces a gradient norm regularization term into the training objective to enhance the robustness of NR IQA models. The regularization term penalizes the $L_1$ norm of the gradient of the model's quality predictions with respect to the input image, thereby encouraging smoother model behavior and reducing vulnerability to adversarial perturbations. Formally, the loss function is modified as follows:

$$\mathcal{L}total = \mathcal{L}_{IQA}(f, x) + \lambda \|\nabla_x f(x)\|_1^2, \tag{11}$$

where $f$ denotes the NR IQA model, $x$ is the input image, $\mathcal{L}_{IQA}(f, x)$ represents the original IQA loss, and $\lambda$ is a hyperparameter balancing the trade-off between task performance and robustness. For our experiments, we set $\lambda = 0.0005$, as suggested in the original paper. The remaining training hyperparameters were aligned with those used for adversarial training.

Both these methods are tailored for IQA task.

**Certified methods**

**Description.** (Cohen et al., 2019) proposed the **Randomized Smoothing (RS)** method to transform any classifier that performs well under Gaussian noise into a new classifier that is certifiably robust to adversarial perturbations under the $l_2$ norm. The overall process of this defense can be described as follows: given an input image, the algorithm samples $N$ noisy variations of this image using a Gaussian noise model with a certain $\sigma$. These images are then passed through the backbone classification model, and the most frequently predicted class is given as the final answer. This approach results in an algorithm that provides a provable answer for the model within a $l_2$ ball. The radius of this ball is calculated based on the difference between the most popular and the second most popular classes across the sampled images used for answer selection. The main disadvantage of the previous approach is that running the classifier on noisy data causes a drop in model accuracy, as it was not trained to handle such data. To address this issue, (Salman et al., 2020) extended randomized smoothing to **Denoised Randomized Smoothing (DRS)** by denoising the noisy image before passing it to the model. Since the noise model is known, training an effective denoiser for a given $\sigma$ is relatively straightforward. (Carlini et al., 2022) extended the approach of (Salman et al., 2020) by replacing the denoiser with a pre-trained denoising diffusion probabilistic model (**Diffusion Denoised Randomized Smoothing (DDRS)**). They used only one diffusion step because it demonstrated high speed and relatively good quality. (Chen et al., 2022b) proposed the **DensePure (DP)** method that involves multiple runs of denoising via the reverse process of the diffusion model (using different random seeds) to generate multiple samples. These samples are then passed through the classifier, and the final prediction is made using majority voting.

(Chiang et al., 2020) proposed a method to certify regression models. Instead of using the most popular class within the $l_2$ ball, they utilize the median of function values. They also theoretically demonstrated that using the median is better than the mean. We denote this method as **Median Smoothing (MS)**. They further extended the method to **Denoised Median Smoothing (DMS)** by adding a denoising step before model prediction to improve accuracy.

**Parameters selection.** Given an input image, the results of the classification-based certified method are the metric score and the certified radius $R$. The method guarantees that the class remains unchanged for the input image within a $l_2$ ball of radius $R$. All classification-based certified methods were run with the following parameters: $\sigma = 0.12, N_0 = 100, N = 1000, \alpha = 0.001$. Here, $\sigma$ is the standard deviation of the Gaussian noise used for sampling, $N_0$ is the number of samples for class selection, $N$ is the number for class certification, and alpha is the probability of class change within the $l_2$ ball of the predicted certified radius $R$.

Given an input image, the results of a regression-based certified method are the metric score and the certified delta. The method guarantees that, within a $l_2$ ball of radius $\epsilon$, the metric score changes by no more than delta. To make this value comparable across metrics, we define the certified relative delta by dividing the certified delta by the metric range. All regression-based certified methods were run with the following parameters: $\sigma = 0.12, \epsilon = 0.05, N = 1000, \alpha = 0.001$.

Scripts for running all these methods are available on GitHub.

Table 9. Experiment to determine the optimal number of classes $N$ for regression metric discretization.

| $N$ | $SROCC_{clear}$ ↑ (no Monte-Carlo sampling) | $Cert.R$ ↑ (with Monte-Carlo sampling) |
|---|---|---|
| 3 | 0.49 | **0.249** |
| 5 | 0.53 | 0.248 |
| 10 | **0.56** | 0.206 |
| 15 | 0.56 | 0.160 |
| 20 | 0.56 | 0.142 |
| ∞ | 0.56 | 0 |

Table 10. Wilcoxon tests in nonadaptive use case of purification defenses on KonIQ dataset for $D_{score}^{(D)}$. Each cell value represents the percentage of experiments in which defense denoted in row statistically performs better in terms of $D_{score}^{(D)}$ than the defense in corresponding column with $p_{value}$=0.05.

| Defense | DiffJPEG | Bilinear Upscale | Unsharp | Resize | Rotate | Crop | Median Blur | JPEG | Gaussian Blur | Color Quantization | DiffPure | Random Noise | Flip | MPRNet | FCN | Real-ESRGAN | DISCO | W/o Defense |
|---|---|---|---|---|---|---|---|---|---|---|---|---|---|---|---|---|---|---|
| DiffJPEG | — | 65.24% | 76.92% | 25.36% | 8.55% | 33.05% | 38.46% | 9.69% | 39.32% | 58.12% | 0.85% | 3.70% | 15.95% | 31.05% | 45.87% | 27.07% | 1.42% | 88.03% |
| Bilinear Upscale | 6.84% | — | 62.39% | 13.39% | 6.27% | 9.97% | 5.70% | 4.84% | 0.00% | 28.77% | 0.00% | 1.99% | 8.26% | 21.08% | 16.81% | 7.12% | 0.00% | 83.48% |
| Unsharp | 0.00% | 1.71% | — | 5.13% | 0.28% | 0.85% | 0.85% | 0.00% | 1.14% | 2.28% | 0.00% | 0.00% | 0.00% | 8.83% | 0.00% | 0.00% | 0.00% | 48.15% |
| Resize | 41.88% | 54.42% | 79.20% | — | 5.70% | 40.17% | 50.43% | 40.17% | 45.87% | 57.83% | 8.26% | 20.23% | 11.68% | 42.74% | 47.58% | 35.33% | 0.00% | 81.20% |
| Rotate | 56.41% | 70.94% | 90.60% | 46.15% | — | 49.29% | 63.53% | 63.53% | 62.68% | 72.36% | 13.96% | 38.75% | 27.92% | 49.00% | 62.11% | 51.00% | 10.26% | 89.17% |
| Crop | 33.05% | 60.40% | 80.91% | 17.38% | 8.55% | — | 44.44% | 36.18% | 42.17% | 58.97% | 10.83% | 29.34% | 11.68% | 37.04% | 49.00% | 41.60% | 10.26% | 86.32% |
| Median Blur | 22.79% | 58.69% | 79.49% | 26.78% | 15.38% | 29.34% | — | 19.09% | 36.18% | 60.97% | 7.98% | 15.10% | 15.95% | 27.35% | 47.29% | 36.18% | 8.83% | 90.03% |
| JPEG | 0.00% | 58.12% | 79.77% | 21.94% | 8.83% | 31.91% | 33.33% | — | 34.47% | 59.54% | 0.85% | 3.99% | 15.10% | 23.65% | 45.58% | 25.64% | 0.85% | 88.89% |
| Gaussian Blur | 19.66% | 79.20% | 75.50% | 25.93% | 11.97% | 23.08% | 26.78% | 16.81% | — | 55.56% | 1.14% | 8.55% | 12.25% | 24.22% | 41.03% | 28.77% | 0.85% | 87.46% |
| Color Quantization | 2.28% | 23.65% | 66.10% | 14.25% | 2.85% | 6.55% | 9.12% | 1.99% | 6.84% | — | 0.28% | 0.00% | 5.41% | 1.14% | 7.69% | 5.70% | 0.57% | 74.64% |
| DiffPure | **80.91%** | **85.75%** | **92.59%** | 68.38% | 47.86% | 60.97% | 80.91% | 84.33% | 75.50% | 86.61% | — | 58.40% | 44.73% | 66.67% | 75.21% | 67.52% | 39.89% | 95.16% |
| Random Noise | 61.54% | 72.93% | 80.06% | 49.29% | 26.50% | 44.16% | 65.53% | 62.39% | 64.67% | 73.22% | 7.98% | — | 28.49% | 52.99% | 60.68% | 46.72% | 9.12% | 82.91% |
| Flip | 35.61% | 51.57% | 66.10% | 34.76% | 8.55% | 36.18% | 45.58% | 34.19% | 43.87% | 54.42% | 9.69% | 22.79% | — | 43.30% | 47.58% | 33.90% | 5.41% | 65.24% |
| MPRNet | 28.49% | 53.28% | 54.13% | 23.93% | 10.26% | 27.64% | 39.89% | 27.07% | 34.47% | 44.16% | 2.85% | 10.26% | 15.38% | — | 39.60% | 25.93% | 2.28% | 62.68% |
| FCN | 9.97% | 42.45% | 82.91% | 18.23% | 4.27% | 7.69% | 21.94% | 9.97% | 24.22% | 39.32% | 1.42% | 5.41% | 4.56% | 24.79% | — | 15.38% | 2.85% | 80.34% |
| Real-ESRGAN | 29.91% | 59.54% | 85.19% | 42.17% | 25.93% | 33.05% | 37.61% | 32.48% | 36.75% | 62.39% | 10.26% | 21.94% | 19.37% | 42.17% | 51.00% | — | 18.80% | 83.19% |
| DISCO | 78.92% | 82.91% | 87.75% | 70.37% | 51.28% | 59.54% | 78.63% | 79.49% | **76.35%** | 81.48% | 23.36% | 60.40% | 43.02% | 65.53% | 72.08% | 54.42% | — | 94.59% |
| W/o Defense | 0.00% | 0.00% | 13.68% | 0.00% | 0.28% | 2.85% | 0.00% | 0.00% | 0.00% | 0.00% | 0.00% | 0.00% | 0.28% | 1.99% | 0.00% | 0.00% | 0.00% | — |

**Classifier-based methods application.** To discretize a regression quality metric for classification-based methods, we divided the metric range into $N$ segments, each corresponding to a specific class. We also added additional classes for metric values that fall below or above the calculated range, ensuring that every metric value is assigned to a class. This resulted in a $(N + 2)$-class metric-classifier. Note that these classes are ordered, with higher class values indicating better quality. Thus, we can measure the quality of the classifier metric in the same way as the regression metric – using relative gain and correlations with subjective scores.

We conducted additional experiments to determine the optimal value of $N$ on PAQ2PIQ NR metric. The main challenge is balancing the trade-off between $SROCC_{clear}$ and $Cert.R$. As the number of classes increases, $SROCC_{clear}$ also increases, but $Cert.R$ decreases. This occurs because a higher number of classes makes it easier to cross class borders during Monte Carlo sampling. Table 9 presents the results of our experiment, indicating that when the number of classes is set to ten, $SROCC_{clear}$ on the discrete metric without Monte Carlo sampling is optimal. Additionally, we measured $Cert.R$ for this number of classes and discovered that $Cert.R$ does not significantly decrease for $N = 10$. Therefore, we chose $N = 10$ to discretize NR metric values in the main experiments of this paper.

### A.4. Statistical tests

We applied the one-sided Wilcoxon Signed Rank Test to assess the statistical significance of defense comparisons, as it is non-parametric and suited for paired samples without assuming normality (which is often the case for attacked IQA models' values) — ideal for adversarial robustness analysis. This test evaluates whether one defense consistently outperforms another in terms of $D_{score}$ and other relevant metrics. Results for different datasets are provided in Tables 10, 15 (non-adaptive) and 16, 17 (adaptive).

Pairwise comparisons yield percentages indicating how often a defense statistically outperforms another under similar adversarial conditions. Stronger defenses (i.e. those yielding high percentages across multiple pairwise comparisons), such as DiffPure and DISCO, show higher effectiveness due to their sophisticated adversarial mitigation techniques, while simpler transformations like blurring or resizing often struggle against stronger attacks.

To ensure reliability, we applied the Bonferroni correction, which controls the family-wise error rate across thousands of

comparisons (17+7 empirical defenses, 14 attacks, 3 intensity scales, and 9 IQA models). This conservative adjustment minimizes false positives, reinforcing the significance of the results.

### A.5. Examples of attacks and defenses

We show examples of attacks and defenses with corresponding metric values in Figure 8. We chose the PAQ2PIQ metric and several types of defenses. The central part of the image is zoomed to show the effects of the defenses and attacks.

We show image artifacts of presented defenses in Figure 9. The attacks were performed on MANIQA metric. We demonstrate that most defenses have artifacts. Most of them include: removing details of the original image (DISCO, MPRNet), altering the image content (Real-ESRGAN, DiffPure), reducing the image clarity (DiffPure, blur defenses), changing image color (FCN), and compression artifacts (JPEG/DiffJPEG, Color Quantization).

### A.6. Additional results

We show performances for evaluated defenses in tables below. Confidence intervals in some table are large due to the fact that we calculate an average score across a large pool of IQA metrics/attacks/datasets. To statistically check what defense is better we provide results of statistical tests A.4.

We analyze how much does $SROCC$ and $PLCC$ correlations are differ in table 20. It reports that ranks in both cases are identical. Thus, in other tables we report only $SROCC$.

Table 11 shows how well defenses can respond to attack of different strengths. In summary, results do not change much across strengths.

Tables 12 and 13 present results of purification defenses on different IQA metrics. We can see that correlations are highly dependent on IQA metric, while top methods in terms of robustness ($D_{score}$, $R_{score}$) are similar across different IQA models.

Table 14 shows scores for different attack types. The experiment showed that undefended images are closer to the original than defended by any defense. The results are the same as on all attack types. JPEG and DiffJPEG show greater $SROCC_{adv}$, while Color Quantization has better PSNR with the original images.

Table 18 reports results for grouped purification defenses. The results demonstrate that compression is one of the best in all metrics except $R_{score}^{(D)}$.

Table 21 reports some score for certified methods. It shows that RS has bigger $Cert.R$ and $Abst.$ among all classification-based defenses, while MS is the best among regression-based defenses.

Figure 10 pictures tradeoffs for Purification-based defenses. It shows the importance of tuning defense parameters for defenses with different parameters.

Figure 11 illustrates difference between $D_{score}$ $D_{score}^{(D)}$ (left) and $R_{score}$ and $R_{score}^{(D)}$ (right). The main finding is that there is high correlation within these pairs of scores with rare exceptions like Random Crop.

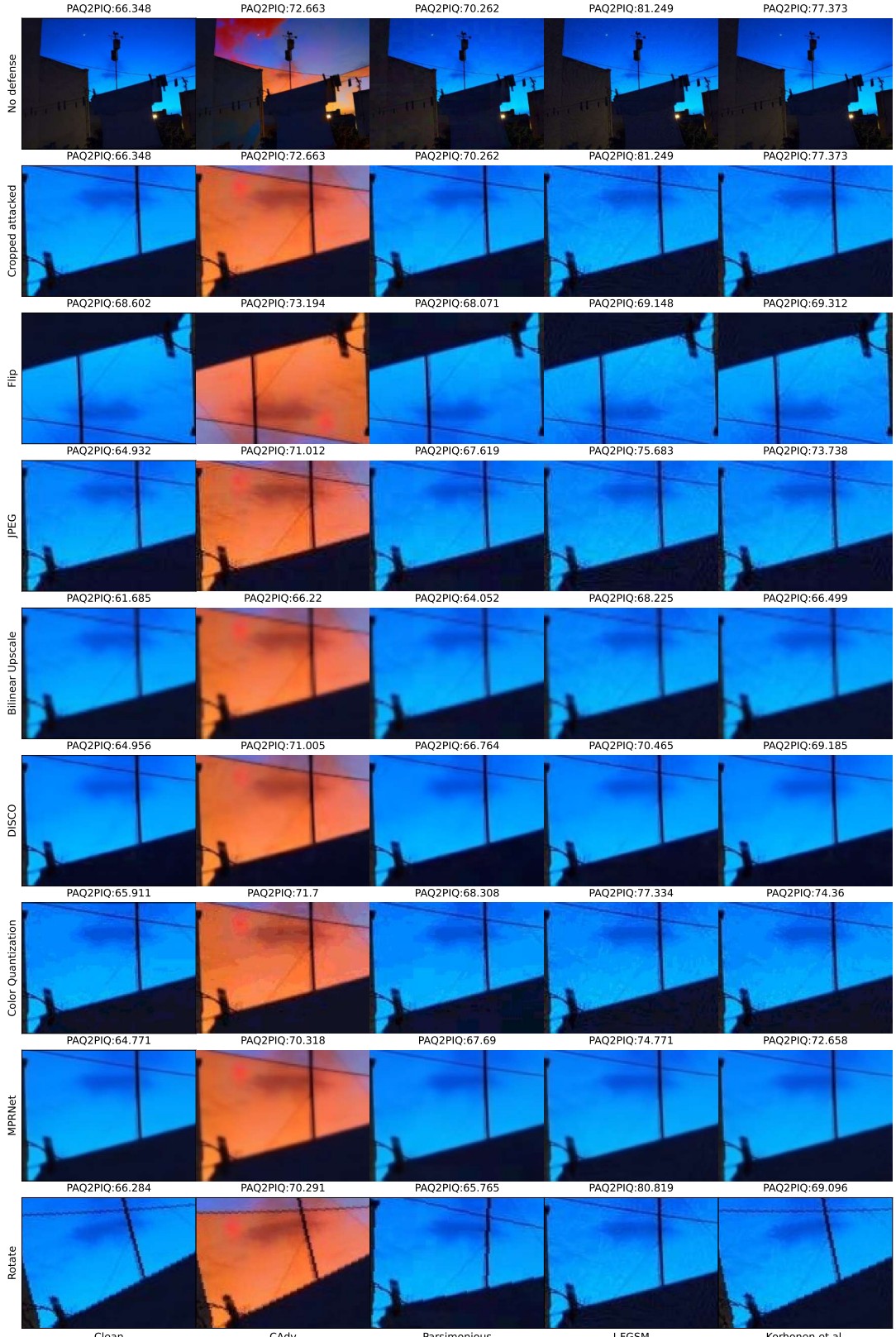

*Figure 8.* Examples of attacks and defenses on PAQ2PIQ metric. The central part of the image is zoomed to show defense effects.

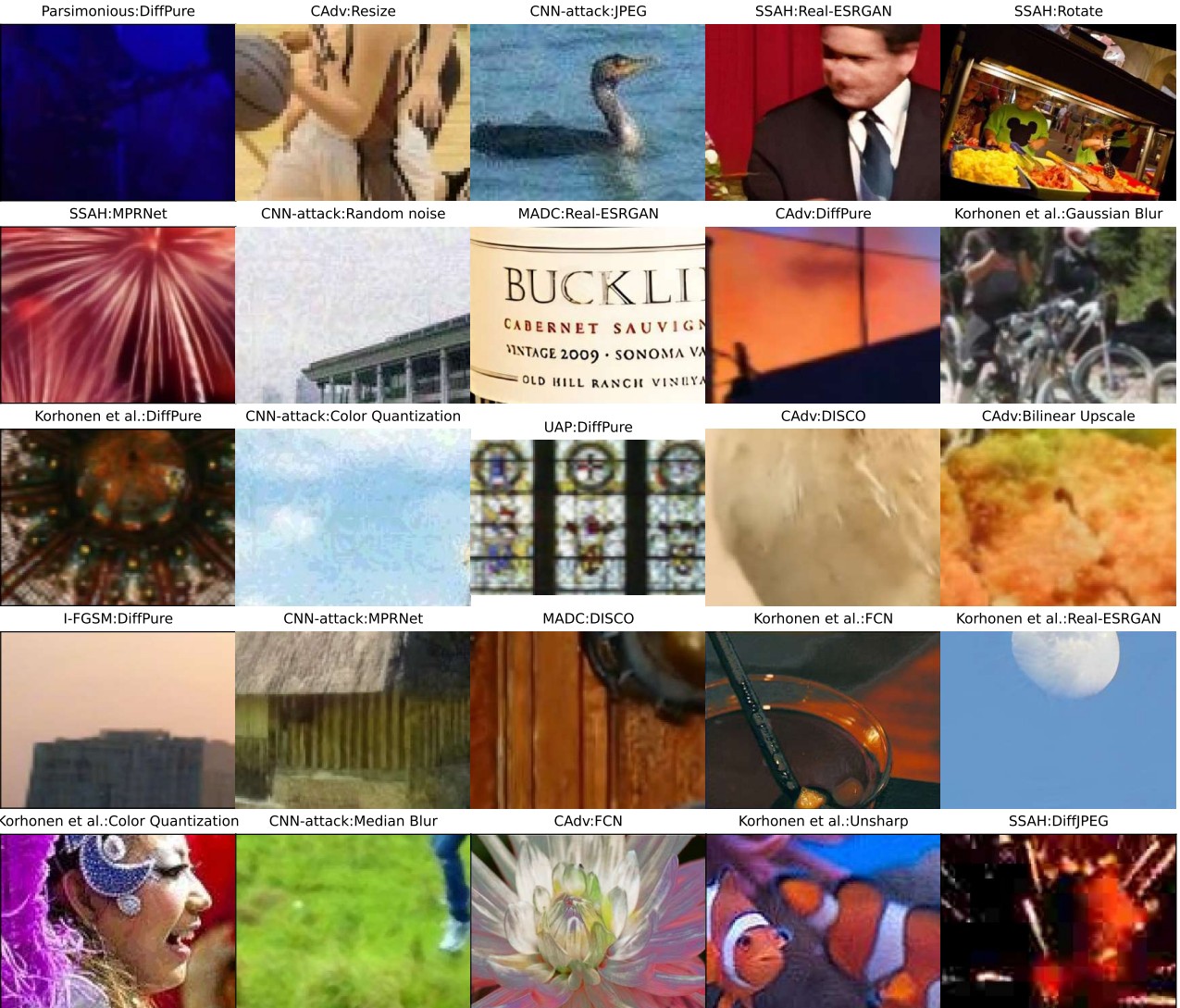

*Figure 9.* Examples of artifacts caused by various defenses when MANIQA metric is attacked. We selectively zoom in on key parts of the images to highlight the details.

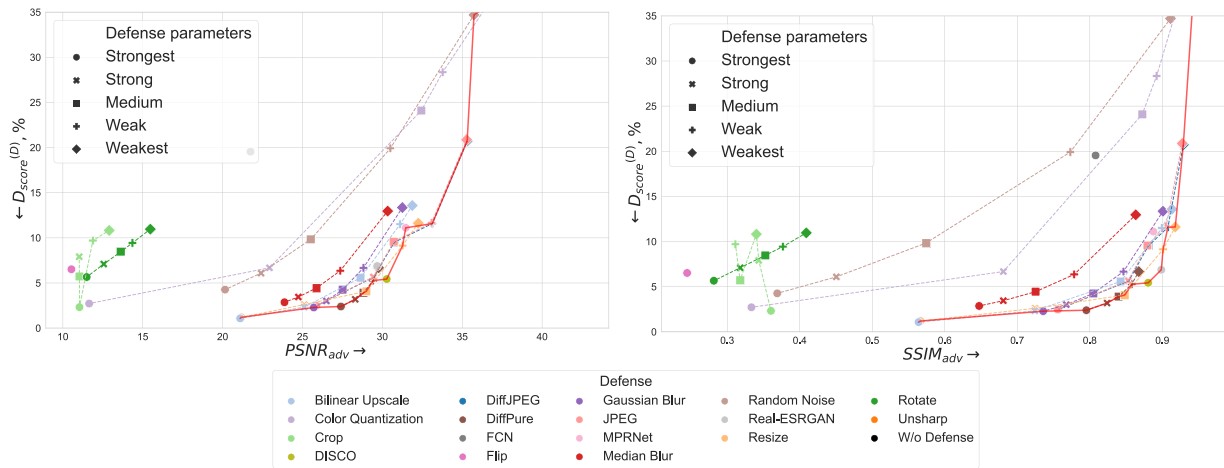

Figure 10. $D_{score}^{(D)}(\downarrow)/PSNR(\uparrow)$ (left) and $D_{score}^{(D)}(\downarrow)/SSIM(\uparrow)$ tradeoffs for Purification-based defenses in non-adaptive scenario averaged across KonIQ-10k, KADID and AGIQA-3K datasets. Red line denotes the Pareto Optimal front.

Table 11. Comparison of purification defenses by different attack strength. Evaluated metrics are averaged across all images, attacks and quality metrics on KonIQ and KADID dataset.

| | Weak | | | Medium | | | Strong | | |
|---|---|---|---|---|---|---|---|---|---|
| | $D_{score}\downarrow$ | $R_{score}\uparrow$ | $SROCC_{adv}\uparrow$ | $D_{score}\downarrow$ | $R_{score}\uparrow$ | $SROCC_{adv}\uparrow$ | $D_{score}\downarrow$ | $R_{score}\uparrow$ | $SROCC_{adv}\uparrow$ |
| W/o Defense | 33.11 / — | 0.822 / — | 0.522 / — | 45.88 / — | 0.639 / — | 0.470 / — | 66.83 / — | 0.502 / — | 0.401 / — |
| Bilinear Upscale | 14.51 / 28.97 | 0.909 / 0.674 | 0.499 / 0.385 | 18.44 / 38.96 | 0.822 / 0.556 | 0.463 / 0.354 | 24.16 / 53.77 | 0.748 / 0.461 | 0.411 / 0.311 |
| Gaussian Blur | 15.10 / 27.56 | 0.878 / 0.655 | 0.465 / 0.421 | 16.52 / 38.85 | 0.836 / 0.518 | 0.437 / 0.371 | 20.13 / 58.06 | 0.761 / 0.361 | 0.400 / 0.256 |
| Resize | 9.99 / 43.42 | 1.118 / 0.525 | 0.594 / 0.411 | 11.78 / 58.27 | 1.053 / 0.404 | 0.582 / 0.347 | 14.15 / 81.16 | 0.995 / 0.270 | 0.561 / 0.247 |
| MPRNet | 12.67 / 30.77 | 0.993 / 0.602 | 0.561 / 0.530 | 14.13 / 42.40 | 0.946 / 0.490 | 0.556 / 0.513 | 16.87 / 60.60 | 0.885 / 0.360 | 0.546 / **0.432** |
| DiffJPEG | 9.71 / 20.72 | 1.138 / 0.774 | **0.634** / 0.573 | 11.96 / 27.29 | 1.057 / 0.660 | **0.632** / **0.555** | 16.89 / 37.29 | 0.967 / 0.547 | **0.619** / **0.491** |
| JPEG | 9.72 / — | 1.139 / — | 0.631 / — | 11.99 / — | 1.054 / — | 0.629 / — | 16.97 / — | 0.963 / — | 0.616 / — |
| Unsharp | 30.91 / 59.45 | 0.652 / 0.388 | 0.484 / 0.403 | 42.38 / 83.78 | 0.530 / 0.232 | 0.419 / 0.304 | 60.27 / 122.51 | 0.439 / 0.097 | 0.376 / 0.248 |
| Median Blur | 13.02 / 34.98 | 0.981 / 0.574 | 0.462 / 0.468 | 15.05 / 47.84 | 0.915 / 0.427 | 0.434 / 0.424 | 18.09 / 67.41 | 0.856 / 0.282 | 0.404 / 0.322 |
| Real-ESRGAN | 21.48 / 26.51 | 0.689 / 0.621 | 0.564 / 0.476 | 23.15 / 31.73 | 0.665 / 0.566 | 0.548 / 0.461 | 26.67 / 41.73 | 0.627 / 0.467 | 0.509 / 0.380 |
| Color Quantization | 15.46 / — | 1.016 / — | 0.586 / — | 21.08 / — | 0.897 / — | 0.568 / — | 36.81 / — | 0.726 / — | 0.499 / — |
| DISCO | 8.72 / 40.31 | 1.176 / 0.514 | 0.607 / 0.479 | 8.30 / 52.05 | 1.193 / 0.438 | 0.612 / 0.453 | **8.29** / 64.85 | **1.190** / 0.406 | 0.613 / 0.429 |
| DiffPure | 17.92 / 15.97 | 0.780 / 0.869 | 0.501 / 0.502 | 17.33 / 20.49 | 0.800 / 0.766 | 0.515 / 0.492 | 17.57 / 32.05 | 0.797 / 0.593 | 0.521 / 0.467 |
| FCN | 15.38 / 47.23 | 0.973 / 0.471 | 0.566 / 0.344 | 20.74 / 67.76 | 0.885 / 0.318 | 0.529 / 0.248 | 30.85 / 100.34 | 0.771 / 0.194 | 0.463 / 0.182 |
| Random Noise | 14.72 / 26.84 | 0.907 / 0.688 | 0.576 / **0.578** | 14.58 / 42.29 | 0.921 / 0.490 | 0.572 / 0.517 | 17.43 / 72.84 | 0.869 / 0.272 | 0.566 / 0.382 |
| Crop | 11.51 / 18.44 | 1.045 / 0.788 | 0.557 / 0.435 | 13.47 / 18.26 | 0.982 / 0.791 | 0.529 / 0.403 | 16.93 / **19.89** | 0.899 / **0.762** | 0.468 / 0.330 |
| Rotate | 9.20 / **10.65** | 1.153 / **1.066** | 0.533 / 0.543 | 9.98 / **15.27** | 1.110 / **0.886** | 0.520 / 0.477 | 11.21 / 23.80 | 1.072 / 0.683 | 0.485 / 0.355 |
| Flip | **6.38** / 47.67 | **1.318** / 0.508 | 0.557 / 0.480 | **7.62** / 67.31 | **1.255** / 0.347 | 0.553 / 0.395 | 9.81 / 99.51 | 1.166 / 0.182 | 0.520 / 0.300 |

Table 12. Per-metric comparison of purification defenses in adaptive use case ($SROCC_{clear}/SROCC_{adv}$). Evaluated metrics are averaged across all images and attacks on KonIQ and KADID dataset.

| Defense | Linearity | KonCept | PAQ2PIQ | MANIQA | Meta-IQA | SPAQ | FPR | TOPIQ(NR) | CLIP-IQA+ |
|---|---|---|---|---|---|---|---|---|---|
| W/o Defense | 0.526 / 0.436 | 0.477 / 0.405 | 0.449 / 0.349 | 0.497 / 0.465 | 0.617 / 0.456 | 0.355 / 0.251 | -0.133 / 0.070 | 0.494 / 0.440 | 0.653 / 0.464 |
| Crop | 0.611 / 0.501 | 0.236 / 0.178 | 0.404 / 0.376 | 0.522 / 0.461 | 0.458 / 0.386 | — / — | 0.173 / 0.154 | 0.611 / 0.540 | 0.592 / 0.518 |
| Real-ESRGAN | 0.613 / 0.461 | **0.708** / 0.544 | 0.510 / 0.414 | **0.786** / 0.616 | 0.278 / 0.303 | 0.438 / 0.343 | 0.295 / **0.265** | 0.576 / 0.506 | 0.681 / 0.495 |
| Unsharp | 0.631 / 0.274 | 0.706 / 0.501 | 0.510 / 0.261 | 0.783 / 0.549 | 0.624 / 0.247 | 0.558 / 0.298 | 0.230 / -0.087 | 0.717 / 0.443 | 0.693 / 0.381 |
| DISCO | 0.662 / 0.554 | 0.519 / 0.486 | 0.555 / 0.423 | 0.630 / 0.583 | **0.643** / **0.523** | 0.571 / 0.407 | 0.108 / -0.075 | 0.719 / 0.655 | 0.653 / 0.530 |
| Resize | 0.676 / 0.372 | 0.481 / 0.328 | 0.510 / 0.314 | 0.565 / 0.490 | 0.575 / 0.331 | — / — | 0.209 / -0.109 | 0.738 / 0.563 | 0.555 / 0.390 |
| Bilinear Upscale | 0.677 / 0.522 | 0.433 / 0.289 | 0.540 / 0.414 | 0.481 / 0.407 | 0.429 / 0.280 | 0.569 / 0.304 | 0.193 / -0.044 | 0.644 / 0.561 | 0.607 / 0.417 |
| DiffPure | 0.680 / 0.643 | 0.515 / 0.499 | 0.564 / **0.483** | 0.558 / 0.584 | 0.424 / 0.404 | 0.577 / 0.467 | 0.061 / 0.149 | 0.611 / 0.577 | 0.685 / 0.576 |
| FCN | 0.700 / 0.306 | 0.624 / 0.376 | 0.546 / 0.207 | 0.684 / 0.450 | 0.595 / 0.167 | 0.515 / 0.222 | 0.190 / -0.141 | 0.675 / 0.396 | 0.693 / 0.340 |
| Gaussian Blur | 0.706 / 0.439 | 0.534 / 0.379 | **0.593** / 0.415 | 0.586 / 0.508 | 0.485 / 0.228 | 0.582 / 0.297 | 0.040 / -0.085 | 0.663 / 0.496 | 0.695 / 0.466 |
| Rotate | 0.713 / 0.484 | 0.674 / 0.546 | 0.557 / 0.420 | 0.731 / 0.710 | 0.506 / 0.268 | 0.582 / 0.406 | 0.274 / 0.228 | 0.734 / 0.594 | 0.700 / 0.470 |
| Median Blur | 0.722 / 0.535 | 0.548 / 0.480 | 0.549 / 0.425 | 0.598 / 0.570 | 0.460 / 0.302 | 0.572 / 0.365 | 0.174 / -0.036 | 0.668 / 0.580 | 0.652 / 0.421 |
| Flip | 0.743 / 0.424 | 0.678 / 0.546 | 0.515 / 0.348 | 0.743 / 0.607 | 0.550 / 0.312 | 0.570 / 0.330 | 0.220 / -0.086 | 0.731 / 0.564 | **0.776** / 0.481 |
| Random Noise | 0.745 / 0.580 | 0.683 / 0.592 | 0.572 / 0.429 | 0.732 / 0.654 | 0.611 / 0.444 | 0.574 / 0.432 | 0.238 / 0.172 | 0.707 / 0.601 | 0.712 / 0.531 |
| DiffJPEG | 0.748 / **0.664** | 0.673 / **0.608** | 0.586 / 0.467 | 0.747 / **0.712** | 0.583 / 0.490 | **0.593** / **0.473** | **0.307** / 0.186 | 0.738 / 0.649 | 0.734 / **0.608** |
| MPRNet | **0.755** / 0.619 | 0.665 / 0.600 | 0.589 / 0.456 | 0.653 / 0.621 | 0.574 / 0.455 | 0.569 / 0.394 | 0.157 / 0.089 | **0.751** / **0.667** | 0.712 / 0.525 |

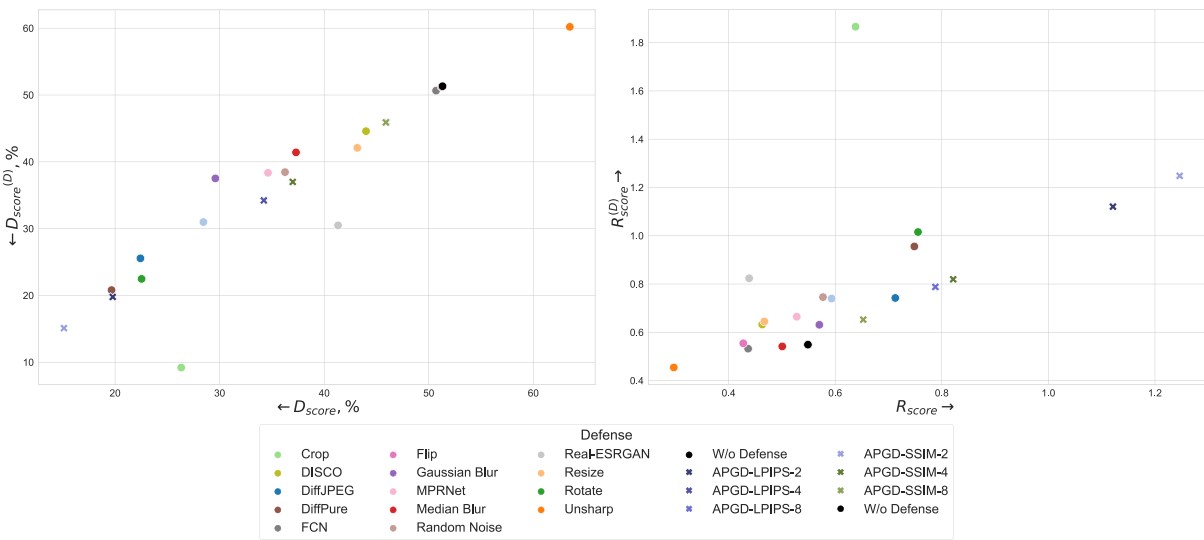

*Figure 11.* Comparison of $D_{score}$ and $D_{score}^{(D)}$ (left) and $R_{score}/R_{score}^{(D)}$ (right) for Purification-based and Adversarial Training defenses in adaptive scenario. Results are averaged across KonIQ-10k, KADID and AGIQA-3K datasets.

*Table 13.* Per-metric comparison of purification defenses in adaptive use case ($D_{score}/R_{score}$). Evaluated metrics are averaged across all images and attacks on KonIQ, KADID and NIPS datasets.

| Defense | Linearity | KonCept | PAQ2PIQ | MANIQA | Meta-IQA | SPAQ | FPR | TOPIQ(NR) | CLIP-IQA+ |
|---|---|---|---|---|---|---|---|---|---|
| W/o Defense | 63.66 / 0.31 | 41.80 / 0.47 | 41.61 / 0.49 | 25.61 / 0.62 | 42.62 / 0.39 | 60.63 / 0.46 | 281.18 / -0.28 | 21.57 / 0.70 | 21.91 / 0.68 |
| Unsharp | 71.57 / 0.21 | 60.29 / 0.23 | 56.56 / 0.26 | 36.70 / 0.41 | 48.48 / 0.26 | 84.57 / 0.20 | 376.46 / -0.45 | 26.08 / 0.56 | 24.13 / 0.60 |
| Resize | 66.23 / 0.25 | 21.52 / 0.67 | 26.15 / 0.64 | 19.13 / 0.68 | 45.11 / 0.29 | — / — | 223.38 / -0.20 | 21.74 / 0.64 | 21.28 / 0.65 |
| Flip | 62.20 / 0.31 | 45.41 / 0.38 | 42.11 / 0.44 | 28.26 / 0.57 | 40.12 / 0.43 | 66.02 / 0.36 | 264.75 / -0.15 | 22.44 / 0.67 | 22.63 / 0.65 |
| FCN | 61.60 / 0.29 | 44.00 / 0.41 | 43.61 / 0.41 | 26.67 / 0.54 | 43.05 / 0.37 | 62.42 / 0.37 | 256.72 / -0.27 | 23.11 / 0.63 | 23.31 / 0.60 |
| DISCO | 56.08 / 0.40 | 30.57 / 0.53 | 33.90 / 0.57 | 24.56 / 0.63 | 36.75 / 0.55 | 57.28 / 0.43 | 151.53 / 0.10 | 20.22 / 0.73 | 20.72 / 0.72 |
| Median Blur | 49.19 / 0.36 | 28.51 / 0.51 | 34.85 / 0.49 | 21.77 / 0.58 | 33.52 / 0.50 | 48.64 / 0.42 | 174.94 / -0.12 | 19.21 / 0.67 | 22.31 / 0.63 |
| MPRNet | 43.85 / 0.42 | 27.66 / 0.54 | 32.94 / 0.52 | 21.32 / 0.61 | 33.94 / 0.52 | 41.87 / 0.49 | 151.72 / -0.04 | 16.67 / 0.73 | 20.77 / 0.66 |
| Random Noise | 42.43 / 0.46 | 34.61 / 0.52 | 37.16 / 0.51 | 24.32 / 0.60 | 35.74 / 0.50 | 45.79 / 0.52 | 165.95 / -0.10 | 17.36 / 0.76 | 18.95 / 0.74 |
| Gaussian Blur | 32.26 / 0.56 | 23.82 / 0.62 | 30.48 / 0.53 | 20.24 / 0.64 | 27.93 / 0.59 | 36.96 / 0.55 | 149.81 / -0.12 | 14.57 / 0.84 | 21.51 / 0.65 |
| Bilinear Upscale | 31.03 / 0.57 | 20.91 / 0.68 | 22.89 / 0.68 | 17.66 / 0.73 | 22.14 / 0.71 | 34.90 / 0.58 | 135.84 / 0.03 | 17.10 / 0.77 | 20.30 / 0.66 |
| Real-ESRGAN | 21.71 / 0.73 | 66.37 / 0.08 | 39.43 / 0.44 | 36.74 / 0.32 | 19.19 / 0.76 | 22.89 / 0.75 | 67.99 / 0.20 | 13.21 / 0.89 | 17.36 / 0.72 |
| Rotate | 21.39 / 0.78 | 23.80 / 0.64 | **13.23 / 0.98** | **9.10 / 1.01** | **14.16 / 0.95** | 26.92 / 0.70 | **12.24 / 0.98** | 10.20 / 1.03 | **10.42 / 1.02** |
| DiffJPEG | 20.07 / 0.74 | 22.80 / 0.64 | 22.70 / 0.66 | 18.04 / 0.71 | 17.54 / 0.79 | 25.02 / 0.70 | 84.61 / 0.15 | 10.94 / 0.95 | 15.59 / 0.81 |
| DiffPure | 19.63 / 0.79 | **17.53 / 0.76** | 19.73 / 0.74 | 14.65 / 0.82 | 14.74 / 0.90 | **21.66 / 0.79** | 66.03 / 0.27 | 13.12 / 0.87 | 13.17 / 0.88 |
| Crop | **16.87 / 0.89** | 31.32 / 0.45 | 20.08 / 0.71 | 15.68 / 0.77 | 19.38 / 0.78 | — / — | 17.24 / 0.85 | **8.17 / 1.14** | 10.96 / 0.96 |

*Table 14.* Comparison of purification defenses by attack type. Evaluated metrics are averaged across all images, attacks and quality metrics for nonadaptive use case on KonIQ and KADID datasets.

| Defense | Restricted WB | | | Unrestricted WB | | | Black-Box | | |
|---|---|---|---|---|---|---|---|---|---|
| | $R_{score}$ ↑ | $SROCC_{adv}$ ↑ | $PSNR$ ↑ | $R_{score}$ ↑ | $SROCC_{adv}$ ↑ | $PSNR$ ↑ | $R_{score}$ ↑ | $SROCC_{adv}$ ↑ | $PSNR$ ↑ |
| W/o Defense | 0.36±0.65 | 0.387±0.29 | **42.12±5.69** | 2.06±1.67 | 0.535±0.26 | **52.31±32.77** | 1.19±0.76 | 0.590±0.31 | 38.87±7.00 |
| Unsharp | 0.30±0.47 | 0.329±0.28 | 30.64±2.20 | 0.89±0.43 | 0.525±0.25 | 26.39±7.31 | 0.87±0.37 | 0.578±0.27 | 28.37±3.96 |
| Real-ESRGAN | 0.62±0.34 | 0.474±0.23 | 31.62±1.37 | 0.70±0.29 | 0.494±0.24 | 25.64±6.49 | 0.75±0.28 | 0.658±0.17 | 28.31±3.86 |
| FCN | 0.62±0.44 | 0.455±0.21 | 20.49±0.33 | 1.07±0.32 | 0.532±0.25 | 18.83±1.87 | 1.21±0.28 | 0.632±0.22 | 21.20±0.46 |
| Color Quantization | 0.63±0.49 | 0.500±0.25 | 33.62±1.63 | 1.08±0.40 | 0.533±0.25 | 27.30±7.77 | 1.21±0.51 | 0.632±0.24 | 32.38±2.45 |
| Bilinear Upscale | 0.68±0.33 | 0.389±0.27 | 33.69±1.94 | 0.92±0.25 | 0.493±0.24 | 27.30±7.99 | 1.01±0.24 | 0.555±0.26 | 29.92±4.70 |
| Gaussian Blur | 0.78±0.24 | 0.374±0.26 | 32.70±1.78 | 0.85±0.22 | 0.443±0.24 | 26.53±7.37 | 0.87±0.27 | 0.522±0.27 | 29.00±4.75 |
| DiffPure | 0.81±0.25 | 0.514±0.19 | 29.46±1.32 | 0.78±0.21 | 0.424±0.22 | 24.39±5.58 | 0.78±0.24 | 0.529±0.22 | 26.26±4.20 |
| Random Noise | 0.82±0.27 | 0.533±0.24 | 26.00±0.69 | 0.83±0.24 | 0.523±0.23 | 22.44±4.03 | 0.99±0.32 | 0.633±0.19 | 25.74±0.65 |
| Crop | 0.83±0.30 | 0.476±0.22 | 11.83±0.13 | 1.05±0.33 | 0.540±0.24 | 11.44±0.50 | 1.19±0.34 | 0.588±0.23 | 11.04±0.74 |
| Median Blur | 0.84±0.28 | 0.385±0.25 | 31.70±1.92 | 0.99±0.31 | 0.455±0.23 | 26.07±7.12 | 1.03±0.28 | 0.503±0.24 | 27.83±5.42 |
| JPEG | 0.91±0.42 | 0.612±0.20 | 33.23±1.72 | 1.14±0.37 | 0.572±0.23 | 26.99±7.34 | 1.23±0.34 | 0.655±0.23 | 30.02±3.37 |
| DiffJPEG | 0.91±0.42 | **0.614±0.20** | 33.29±1.72 | 1.14±0.37 | **0.575±0.23** | 27.02±7.36 | 1.22±0.33 | **0.660±0.22** | 30.03±3.39 |
| MPRNet | 0.94±0.28 | 0.557±0.16 | 32.36±1.61 | 0.98±0.26 | 0.511±0.20 | 26.14±7.20 | 0.99±0.26 | 0.629±0.18 | 27.93±4.92 |
| Resize | 0.94±0.35 | 0.545±0.21 | 32.65±1.78 | 1.08±0.33 | 0.555±0.22 | 26.55±7.39 | 1.23±0.27 | 0.639±0.21 | 28.96±4.73 |
| Rotate | 1.08±0.20 | 0.486±0.20 | 11.41±0.38 | 1.11±0.24 | 0.513±0.23 | 11.09±0.70 | 1.18±0.21 | 0.558±0.24 | 10.87±1.26 |
| Flip | 1.11±0.28 | 0.564±0.23 | 10.85±0.21 | 1.29±0.27 | 0.553±0.27 | 10.53±0.41 | **1.43±0.25** | 0.655±0.21 | 10.21±0.52 |
| DISCO | **1.20±0.22** | 0.594±0.22 | 29.62±1.29 | 1.07±0.27 | 0.544±0.22 | 24.21±5.35 | 1.20±0.25 | 0.651±0.20 | 26.64±3.57 |

*Table 15.* Wilcoxon tests in nonadaptive use case of purification defenses on NIPS dataset for $SSIM$ scores.

| Defense | DiffJPEG | Bilinear Upscale | Unsharp | Resize | Rotate | Crop | Median Blur | JPEG | Gaussian Blur | Color Quantization | DiffPure | Random Noise | Flip | MPRNet | FCN | Real-ESRGAN | DISCO | W/o Defense |
|---|---|---|---|---|---|---|---|---|---|---|---|---|---|---|---|---|---|
| DiffJPEG | — | 7.12% | 39.03% | 7.41% | **100.00%** | **100.00%** | 63.25% | 33.33% | 7.69% | 62.11% | 65.53% | **100.00%** | **100.00%** | 0.00% | 17.66% | 2.28% | 3.13% | 1.99% |
| Bilinear Upscale | 52.71% | — | 44.16% | 37.04% | 100.00% | 100.00% | 79.77% | 53.28% | 43.59% | 62.11% | 82.05% | 100.00% | 100.00% | 0.00% | 53.85% | 1.42% | 0.00% | 4.84% |
| Unsharp | 29.91% | 10.54% | — | 16.24% | 100.00% | 100.00% | 54.99% | 32.76% | 19.37% | 45.58% | 60.68% | 98.58% | 100.00% | 2.28% | 24.22% | 3.42% | 5.41% | 0.00% |
| Resize | 19.66% | 0.00% | 35.04% | — | 88.89% | 88.89% | 63.53% | 35.04% | 21.94% | 49.29% | 53.56% | 88.89% | 88.89% | 0.00% | 45.30% | 0.57% | 0.00% | 3.99% |
| Rotate | 0.00% | 0.00% | 0.00% | 0.00% | — | 0.00% | 0.00% | 0.00% | 0.00% | 0.00% | 0.00% | 0.00% | 62.39% | 0.00% | 0.00% | 0.00% | 0.00% | 0.00% |
| Crop | 0.00% | 0.00% | 0.00% | 0.00% | 86.32% | — | 0.00% | 0.00% | 0.00% | 0.00% | 0.00% | 0.00% | 100.00% | 0.00% | 0.00% | 0.00% | 0.00% | 0.00% |
| Median Blur | 0.00% | 0.00% | 21.94% | 0.00% | 100.00% | 100.00% | — | 0.00% | 0.00% | 2.28% | 0.00% | 100.00% | 100.00% | 0.00% | 0.00% | 0.00% | 0.00% | 0.00% |
| JPEG | 0.00% | 6.27% | 36.18% | 7.12% | 100.00% | 100.00% | 62.68% | — | 7.69% | 60.40% | 62.39% | 100.00% | 100.00% | 0.00% | 11.40% | 0.85% | 1.99% | 1.71% |
| Gaussian Blur | 18.80% | 0.00% | 37.89% | 8.83% | 100.00% | 100.00% | 65.81% | 27.64% | — | 55.84% | 63.53% | 100.00% | 100.00% | 0.00% | 45.01% | 0.57% | 0.00% | 4.56% |
| Color Quantization | 0.00% | 0.00% | 21.08% | 0.00% | 100.00% | 100.00% | 14.25% | 0.00% | 0.00% | — | 16.52% | 100.00% | 100.00% | 0.00% | 0.00% | 0.00% | 0.00% | 0.00% |
| DiffPure | 0.57% | 0.00% | 23.93% | 0.00% | 100.00% | 100.00% | 17.66% | 0.85% | 0.00% | 17.09% | — | 100.00% | 100.00% | 0.00% | 1.71% | 0.00% | 0.00% | 1.14% |
| Random Noise | 0.00% | 0.00% | 0.00% | 0.00% | 99.43% | 84.62% | 0.00% | 0.00% | 0.00% | 0.00% | 0.00% | — | 100.00% | 0.00% | 0.00% | 0.00% | 0.00% | 0.00% |
| Flip | 0.00% | 0.00% | 0.00% | 0.00% | 0.00% | 0.00% | 0.00% | 0.00% | 0.00% | 0.00% | 0.00% | 0.00% | — | 0.00% | 0.00% | 0.00% | 0.00% | 0.00% |
| MPRNet | 52.71% | 16.24% | 42.74% | 41.31% | 100.00% | 100.00% | 63.53% | 54.13% | 45.87% | 61.54% | 62.96% | 100.00% | 100.00% | — | 54.13% | 9.40% | 2.85% | 5.41% |
| FCN | 1.14% | 4.56% | 29.06% | 4.84% | 100.00% | 100.00% | 51.28% | 1.42% | 5.98% | 36.18% | 42.45% | 100.00% | 100.00% | 0.85% | — | 2.28% | 1.42% | 0.85% |
| Real-ESRGAN | 72.08% | 9.97% | 47.86% | 51.00% | 100.00% | 100.00% | 74.93% | 61.54% | 59.83% | 64.67% | 81.20% | 100.00% | 100.00% | 1.99% | 53.85% | — | 1.14% | 4.27% |
| DISCO | 71.23% | 54.70% | 62.96% | 63.25% | 100.00% | 100.00% | 81.20% | 72.36% | 72.36% | 88.03% | **95.73%** | 100.00% | 100.00% | 45.87% | 67.24% | 48.72% | — | **18.52%** |
| W/o Defense | **90.03%** | **88.03%** | **100.00%** | **81.48%** | 100.00% | 100.00% | **97.44%** | **90.03%** | **91.17%** | **95.44%** | 92.88% | 100.00% | 100.00% | **85.19%** | **87.75%** | **86.04%** | 65.81% | — |

*Table 16.* Wilcoxon tests in adaptive use case of purification-based and Adversarial Training defenses on AGIQA dataset and **Linearity**, **Koncept** IQA metrics for $R_{score}$ values.

| | FCN | MPRNet | Median Blur | DISCO | Bilinear Upscale | Flip | DiffPure | Crop | DiffJPEG | Real-ESRGAN | Gaussian Blur | Resize | Unsharp | Rotate | Random Noise | W/o Defense | APGD-LPIPS-2 | APGD-LPIPS-4 | APGD-LPIPS-8 | APGD-SSIM-2 | APGD-SSIM-4 | APGD-SSIM-8 |
|---|---|---|---|---|---|---|---|---|---|---|---|---|---|---|---|---|---|---|---|---|---|---|
| FCN | — | 10.42% | 12.50% | 50.00% | 16.67% | 39.58% | 18.75% | 22.92% | 4.17% | 16.67% | 14.58% | 47.92% | 87.50% | 6.25% | 10.42% | 39.58% | 2.08% | 4.17% | 41.67% | 0.00% | 10.42% | 25.00% |
| MPRNet | 87.50% | — | 77.08% | 58.33% | 27.08% | 89.58% | 25.00% | 39.58% | 27.08% | 45.83% | 22.92% | 64.58% | 87.50% | 14.58% | 50.00% | 83.33% | 6.25% | 41.67% | 43.75% | 4.17% | 43.75% | 54.17% |
| Median Blur | 83.33% | 16.67% | — | 62.50% | 25.00% | 83.33% | 22.92% | 39.58% | 22.92% | 43.75% | 27.08% | 62.50% | 87.50% | 16.67% | 45.83% | 83.33% | 8.33% | 31.25% | 47.92% | 4.17% | 39.58% | 45.83% |
| DISCO | 39.58% | 35.42% | — | — | 25.00% | 31.25% | 45.83% | 31.25% | 20.83% | 37.50% | 50.00% | 43.75% | 12.50% | 41.67% | 45.83% | 6.25% | 41.67% | 45.83% | — | — | — | — |
| Bilinear Upscale | 81.25% | 62.50% | 68.75% | 68.75% | — | 79.17% | 14.58% | 56.25% | 39.58% | 47.92% | 37.50% | 58.33% | 81.25% | 35.42% | 47.92% | 81.25% | 33.33% | 37.50% | 50.00% | 10.42% | 37.50% | 54.17% |
| Flip | 39.58% | 8.33% | 12.50% | 54.17% | 18.75% | — | 4.17% | 16.67% | 18.75% | 25.00% | 4.17% | 87.50% | 87.50% | 4.17% | 2.08% | 68.75% | 2.08% | 4.17% | 41.67% | 0.00% | 10.42% | 31.25% |
| DiffPure | 79.17% | 72.92% | 75.00% | 66.67% | 77.08% | 79.17% | — | 64.58% | 66.67% | 66.67% | 64.58% | 72.92% | 83.33% | 72.92% | 75.00% | 81.25% | 66.67% | 72.92% | 77.08% | 37.50% | 72.92% | 77.08% |
| Crop | 72.92% | 52.08% | 58.33% | 47.92% | 37.50% | 70.83% | 16.67% | — | 47.92% | 60.42% | 43.75% | 45.83% | 45.83% | 47.92% | 58.33% | 72.92% | 33.33% | 58.33% | 72.92% | 18.75% | 60.42% | 68.75% |
| DiffJPEG | 89.58% | 62.50% | 68.75% | 60.42% | 58.33% | 93.75% | 33.33% | 41.67% | — | 60.42% | 62.50% | 66.67% | 89.58% | 29.17% | 66.67% | 87.50% | 18.75% | 58.33% | 83.33% | 10.42% | 62.50% | 81.25% |
| Real-ESRGAN | 70.83% | 47.92% | 47.92% | 43.75% | 43.75% | 66.67% | 14.58% | 18.75% | 31.25% | — | 43.75% | 50.00% | 83.33% | 37.50% | 43.75% | 72.92% | 27.08% | 39.58% | 79.17% | 2.08% | 54.17% | 68.75% |
| Gaussian Blur | 81.25% | 75.00% | 68.75% | 68.75% | 56.25% | 81.25% | 33.33% | 43.75% | 35.42% | 45.83% | — | 68.75% | 83.33% | 29.17% | 64.58% | 81.25% | 20.83% | 43.75% | 77.08% | 8.33% | 43.75% | 79.17% |
| Resize | 47.92% | 35.42% | 37.50% | 54.17% | 35.42% | 50.00% | 16.67% | 45.83% | 33.33% | 39.58% | 27.08% | — | 66.67% | 37.50% | 52.08% | 29.17% | 35.42% | 35.42% | 37.50% | 6.25% | 35.42% | 37.50% |
| Unsharp | 4.17% | 12.50% | 12.50% | 31.25% | 12.50% | 12.50% | 16.67% | 22.92% | 4.17% | 10.42% | 14.58% | 27.08% | — | 6.25% | 10.42% | 0.00% | 2.08% | 4.17% | 31.25% | 0.00% | 4.17% | 10.42% |
| Rotate | 89.58% | 83.33% | 79.17% | 75.00% | 58.33% | 89.58% | 22.92% | 43.75% | 45.83% | 91.67% | — | 83.33% | 83.33% | — | 79.17% | 83.33% | 6.25% | 16.67% | 75.00% | 0.00% | 75.00% | 81.25% |
| Random Noise | 85.42% | 43.75% | 54.17% | 58.33% | 41.67% | 97.92% | 22.92% | 37.50% | 29.17% | 43.75% | 29.17% | 60.42% | 89.58% | 12.50% | — | 83.33% | 6.25% | 16.67% | 64.58% | 4.17% | 43.75% | 70.83% |
| W/o Defense | 47.92% | 16.67% | 16.67% | 52.08% | 18.75% | 20.83% | 18.75% | 22.92% | 4.17% | 10.42% | 18.75% | 47.92% | 93.75% | 16.67% | 12.50% | — | 6.25% | 8.33% | 41.67% | 0.00% | 14.58% | 31.25% |
| APGD-LPIPS-2 | 93.75% | 81.25% | 85.42% | 70.83% | 60.42% | 93.75% | 27.08% | 54.17% | 81.25% | 58.33% | 66.67% | 64.58% | 95.83% | 62.50% | 89.58% | 89.58% | — | 87.50% | 89.58% | 0.00% | 85.42% | 87.50% |
| APGD-LPIPS-4 | 87.50% | 56.25% | 58.33% | 56.25% | 52.08% | 89.58% | 25.00% | 50.00% | 50.00% | 50.00% | 50.00% | 56.25% | 64.58% | 16.67% | 16.67% | 83.33% | 6.25% | — | 41.67% | 0.00% | 41.67% | 81.25% |
| APGD-LPIPS-8 | 54.17% | 41.67% | 47.92% | 50.00% | 22.92% | 58.33% | 18.75% | 22.92% | 14.58% | 10.42% | 12.50% | 56.25% | 64.58% | 14.58% | 20.83% | 52.08% | 4.17% | 8.33% | — | 0.00% | 4.17% | 14.58% |
| APGD-SSIM-2 | **100.00%** | **91.67%** | **93.75%** | **93.75%** | 79.17% | **96.25%** | 54.17% | **68.75%** | 75.00% | **89.58%** | 81.25% | **100.00%** | **93.75%** | **100.00%** | 89.58% | **100.00%** | 89.58% | 20.83% | 75.00% | — | **91.67%** | **93.75%** |
| APGD-SSIM-4 | 75.00% | 52.08% | 56.25% | 56.25% | 50.00% | 72.92% | 25.00% | 33.33% | 27.08% | 27.08% | 50.00% | 64.58% | 91.67% | 16.67% | 50.00% | 68.75% | 8.33% | 18.75% | 81.25% | 6.25% | — | 79.17% |
| APGD-SSIM-8 | 62.50% | 41.67% | 41.67% | 52.08% | 22.92% | 58.33% | 18.75% | 27.08% | 12.50% | 16.67% | 14.58% | 60.42% | 87.50% | 14.58% | 18.75% | 54.17% | 6.25% | 14.58% | 45.83% | 0.00% | 6.25% | — |

*Table 17.* Wilcoxon tests in adaptive use case of purification-based and Adversarial Training defenses on NIPS dataset and **Linearity**, **Koncept** IQA metrics for $PSNR$ values.

| | FCN | MPRNet | Median Blur | DISCO | Bilinear Upscale | Flip | DiffPure | Crop | DiffJPEG | Real-ESRGAN | Gaussian Blur | Resize | Unsharp | Rotate | Random Noise | W/o Defense | APGD-LPIPS-2 | APGD-LPIPS-4 | APGD-LPIPS-8 | APGD-SSIM-2 | APGD-SSIM-4 | APGD-SSIM-8 |
|---|---|---|---|---|---|---|---|---|---|---|---|---|---|---|---|---|---|---|---|---|---|---|
| FCN | — | 0.00% | 0.00% | 0.00% | 2.08% | **100.00%** | 0.00% | **100.00%** | 0.00% | **0.00%** | 0.00% | 2.08% | 0.00% | **100.00%** | 0.00% | 0.00% | 2.08% | 2.08% | 0.00% | 4.17% | 0.00% | 0.00% |
| MPRNet | 97.92% | — | 93.75% | 0.00% | **100.00%** | 100.00% | 95.83% | 100.00% | 89.58% | 0.00% | 91.67% | **100.00%** | **100.00%** | **100.00%** | 0.00% | 0.00% | 4.17% | 2.08% | 2.08% | 6.25% | 2.08% | 2.08% |
| Median Blur | 97.92% | 0.00% | — | 0.00% | 100.00% | 100.00% | 100.00% | 2.08% | 100.00% | 0.00% | 100.00% | 100.00% | 100.00% | 100.00% | 0.00% | 0.00% | 4.17% | 2.08% | 2.08% | 4.17% | 2.08% | 2.08% |
| DISCO | **100.00%** | **100.00%** | **100.00%** | — | 100.00% | 100.00% | **100.00%** | 100.00% | **100.00%** | 0.00% | **100.00%** | 100.00% | 100.00% | 100.00% | 25.00% | 14.58% | 20.83% | 16.67% | 16.67% | 22.92% | 20.83% | 16.67% |
| Bilinear Upscale | 93.75% | 0.00% | 0.00% | 0.00% | — | 100.00% | 0.00% | 100.00% | 2.08% | 0.00% | 0.00% | 39.58% | 4.17% | 100.00% | 0.00% | 0.00% | 2.08% | 2.08% | 0.00% | 2.08% | 0.00% | 0.00% |
| Flip | 0.00% | 0.00% | 0.00% | 0.00% | 0.00% | — | 0.00% | 0.00% | 0.00% | 0.00% | 0.00% | 0.00% | 0.00% | 0.00% | 0.00% | 0.00% | 0.00% | 0.00% | 0.00% | 0.00% | 0.00% | 0.00% |
| DiffPure | 97.92% | 0.00% | 93.75% | 0.00% | 100.00% | 100.00% | — | 100.00% | 89.58% | 0.00% | 91.67% | 100.00% | 97.92% | 100.00% | 0.00% | 0.00% | 4.17% | 2.08% | 2.08% | 6.25% | 2.08% | 2.08% |
| Crop | 0.00% | 0.00% | 0.00% | 0.00% | 0.00% | 100.00% | 0.00% | — | 0.00% | 0.00% | 0.00% | 0.00% | 0.00% | 0.00% | 0.00% | 0.00% | 0.00% | 0.00% | 0.00% | 0.00% | 0.00% | 0.00% |
| DiffJPEG | 95.83% | 2.08% | 91.67% | 0.00% | 97.92% | 100.00% | 6.25% | 100.00% | — | 0.00% | 6.25% | 97.92% | 95.83% | 100.00% | 0.00% | 0.00% | 4.17% | 0.00% | 0.00% | 8.33% | 0.00% | 0.00% |
| Real-ESRGAN | 0.00% | 0.00% | 0.00% | 0.00% | 0.00% | 0.00% | 0.00% | 0.00% | 0.00% | — | 0.00% | 0.00% | 0.00% | 0.00% | 0.00% | 0.00% | 0.00% | 0.00% | 0.00% | 0.00% | 0.00% | 0.00% |
| Gaussian Blur | 97.92% | 0.00% | 95.83% | 0.00% | 100.00% | 100.00% | 4.17% | 100.00% | 22.92% | 0.00% | — | 100.00% | 100.00% | 100.00% | 0.00% | 0.00% | 4.17% | 2.08% | 2.08% | 6.25% | 2.08% | 2.08% |
| Resize | 91.67% | 0.00% | 0.00% | 45.83% | 100.00% | 100.00% | 0.00% | 100.00% | 0.00% | 0.00% | 0.00% | — | 6.25% | 100.00% | 0.00% | 0.00% | 2.08% | 0.00% | 0.00% | 2.08% | 0.00% | 0.00% |
| Unsharp | 93.75% | 0.00% | 0.00% | 0.00% | 91.67% | 100.00% | 0.00% | 100.00% | 2.08% | 0.00% | 0.00% | 91.67% | — | 100.00% | 0.00% | 0.00% | 2.08% | 2.08% | 0.00% | 4.17% | 2.08% | 0.00% |
| Rotate | 0.00% | 0.00% | 0.00% | 0.00% | 0.00% | 100.00% | 0.00% | 0.00% | 0.00% | 0.00% | 0.00% | 0.00% | 0.00% | — | 0.00% | 0.00% | 0.00% | 0.00% | 0.00% | 0.00% | 0.00% | 0.00% |
| Random Noise | 97.92% | 97.92% | 100.00% | 75.00% | 100.00% | 100.00% | 100.00% | 100.00% | 100.00% | 0.00% | 100.00% | 100.00% | 100.00% | 100.00% | — | 0.00% | 6.25% | 6.25% | 12.50% | 4.17% | 2.08% | 2.08% |
| W/o Defense | 100.00% | 100.00% | 100.00% | 81.25% | 100.00% | 100.00% | 100.00% | 100.00% | 97.92% | 0.00% | 100.00% | 100.00% | 100.00% | 100.00% | 97.92% | — | 35.42% | 35.42% | **41.67%** | **50.00%** | 37.50% | 41.67% |
| APGD-LPIPS-2 | 95.83% | 87.50% | 91.67% | 72.92% | 95.83% | 100.00% | 89.58% | 100.00% | 87.50% | 0.00% | 89.58% | 95.83% | 93.75% | 100.00% | 85.42% | 47.92% | — | **50.00%** | 31.25% | 35.42% | **50.00%** | **50.00%** |
| APGD-LPIPS-4 | 95.83% | 89.58% | 93.75% | 75.00% | 97.92% | 100.00% | 89.58% | 100.00% | 87.50% | 0.00% | 89.58% | 97.92% | 93.75% | 100.00% | 87.50% | 27.08% | 39.58% | — | 31.25% | 35.42% | 20.83% | 25.00% |
| APGD-LPIPS-8 | 95.83% | 87.50% | 93.75% | 72.92% | 97.92% | 100.00% | 89.58% | 100.00% | 89.58% | 0.00% | 89.58% | 97.92% | 97.92% | 100.00% | 87.50% | 47.92% | 43.75% | 47.92% | — | **66.67%** | 50.00% | 43.75% |
| APGD-SSIM-2 | 95.83% | 87.50% | 89.58% | 70.83% | 91.67% | 100.00% | 87.50% | 100.00% | 87.50% | 0.00% | 89.58% | 91.67% | 91.67% | 100.00% | 85.42% | 45.83% | 29.17% | 50.00% | 20.83% | — | 41.67% | 43.75% |
| APGD-SSIM-4 | 95.83% | 87.50% | 91.67% | 75.00% | 95.83% | 100.00% | 89.58% | 100.00% | 87.50% | 0.00% | 89.58% | 97.92% | 93.75% | 100.00% | 87.50% | 29.17% | 29.17% | 18.75% | 25.00% | 41.67% | — | 31.25% |
| APGD-SSIM-8 | 97.92% | 89.58% | 93.75% | 72.92% | 97.92% | 100.00% | 91.67% | 100.00% | 89.58% | 0.00% | 91.67% | 97.92% | 97.92% | 100.00% | 87.50% | 22.92% | 39.58% | 33.33% | 8.33% | 45.83% | 22.92% | — |

*Table 18.* Comparison of defenses by defense type. Evaluated metrics are averaged across all images, attacks and quality metrics for nonadaptive/adaptive use cases on KonIQ and KADID datasets.

| Defense | $D_{score}^{(D)} \downarrow$ | $D_{score} \downarrow$ | $R_{score}^{(D)} \uparrow$ | $R_{score} \uparrow$ | $SROCC_{adv} \uparrow$ | $SROCC_{clear} \uparrow$ | $PSNR \uparrow$ |
|---|---|---|---|---|---|---|---|
| Filtering | 21.13 / 27.17 | 20.39 / 22.34 | 0.63 / 0.49 | 0.72 / 0.68 | 0.499 / 0.545 | 0.631 / 0.628 | 19.53 / 20.14 |
| Compression | 21.86 / **15.60** | **18.20** / **11.29** | 0.65 / 0.75 | **0.81** / **0.99** | 0.561 / **0.635** | **0.687** / **0.697** | **19.96** / 20.46 |
| Spatial Transforms | 21.20 / 29.95 | 20.27 / 26.53 | 0.64 / 0.46 | 0.69 / 0.62 | **0.578** / 0.508 | 0.684 / 0.604 | 19.62 / 16.57 |
| Denoising | 17.26 / 19.90 | 26.05 / 25.95 | 0.80 / 0.71 | 0.59 / 0.60 | 0.533 / 0.569 | 0.664 / 0.672 | 19.66 / 20.09 |
| With Randomness | 14.93 / 16.71 | 19.17 / 22.29 | 0.83 / **0.84** | 0.77 / 0.69 | 0.523 / 0.528 | 0.634 / 0.596 | 18.81 / 14.81 |
| Adv. Defenses | **8.15** / 26.86 | 23.14 / 22.35 | **1.11** / 0.50 | 0.63 / 0.69 | 0.474 / 0.538 | 0.583 / 0.626 | 19.09 / 19.16 |
| Adv. Training | — / 22.41 | — / 22.41 | — / 0.68 | — / 0.68 | — / 0.552 | — / 0.667 | — / — |

*Table 19.* Comparison of purification defenses by dataset. Evaluated metrics are averaged across all images, attacks and quality metrics for nonadaptive/adaptive use cases.

| | KonIQA1K | | | KADID1K | | | AGIQA-3K | | | NIPS | |
|---|---|---|---|---|---|---|---|---|---|---|---|
| | $D_{score} \downarrow$ | $R_{score} \uparrow$ | $SROCC_{clear} \uparrow$ | $D_{score} \downarrow$ | $R_{score} \uparrow$ | $SROCC_{clear} \uparrow$ | $D_{score} \downarrow$ | $R_{score} \uparrow$ | $SROCC_{clear} \uparrow$ | $D_{score} \downarrow$ | $R_{score} \uparrow$ |
| W/o Defense | 51.32 / — | 0.57 / — | **0.778** / — | 45.90 / — | 0.74 / — | 0.487 / — | 55.89 / — | 0.57 / — | 0.586 / — | 47.73 / — | 0.60 / — |
| Unsharp | 47.09 / 92.16 | 0.48 / 0.21 | 0.767 / 0.766 | 41.96 / 84.67 | 0.60 / 0.27 | 0.462 / 0.423 | 38.49 / 102.72 | 0.55 / 0.16 | **0.625** / 0.596 | 45.11 / 85.18 | 0.50 / 0.28 |
| Color Quantization | 24.43 / — | 0.83 / — | 0.760 / — | 24.47 / — | 0.93 / — | 0.475 / — | 24.67 / — | 0.86 / — | 0.546 / — | 25.36 / — | 0.85 / — |
| Bilinear Upscale | 19.66 / 45.22 | 0.82 / 0.52 | 0.679 / 0.587 | 18.42 / 36.11 | 0.84 / 0.61 | 0.512 / 0.420 | 12.59 / 47.36 | 1.02 / 0.50 | 0.542 / 0.432 | 20.83 / 26.88 | 0.68 / 0.68 |
| FCN | 22.93 / 73.59 | 0.84 / 0.31 | 0.733 / 0.746 | 21.72 / 70.03 | 0.92 / 0.34 | 0.465 / 0.391 | 26.29 / 74.13 | 0.79 / 0.26 | 0.541 / 0.548 | 18.38 / 50.78 | 0.91 / 0.46 |
| Gaussian Blur | 16.80 / 42.72 | 0.83 / 0.49 | 0.607 / 0.615 | 17.70 / 40.12 | 0.82 / 0.53 | 0.512 / 0.461 | 16.45 / 50.24 | 0.90 / 0.42 | 0.568 / 0.490 | 15.83 / 36.36 | 0.82 / 0.60 |
| Median Blur | 15.17 / 51.43 | 0.93 / 0.41 | 0.668 / 0.678 | 15.60 / 48.65 | 0.90 / 0.45 | 0.440 / 0.402 | 15.98 / 57.85 | 0.95 / 0.36 | 0.571 / 0.512 | 11.82 / 44.05 | 0.98 / 0.49 |
| Real-ESRGAN | 26.36 / 33.41 | 0.59 / 0.54 | 0.719 / 0.682 | 21.18 / 33.15 | 0.73 / 0.57 | 0.484 / 0.385 | 18.49 / 30.48 | 0.61 / 0.57 | 0.464 / 0.457 | 25.38 / 35.13 | 0.61 / 0.53 |
| JPEG | 13.34 / — | 1.02 / — | 0.767 / — | 12.45 / — | 1.08 / — | 0.530 / — | 12.97 / — | 1.08 / — | 0.593 / — | 10.35 / — | 1.06 / — |
| DiffJPEG | 13.27 / 28.33 | 1.02 / 0.65 | 0.770 / 0.765 | 12.43 / 28.46 | 1.09 / 0.67 | 0.532 / 0.484 | 12.96 / 34.54 | 1.08 / 0.57 | 0.593 / 0.579 | 10.33 / 22.22 | 1.06 / 0.73 |
| Resize | 12.35 / 67.49 | 1.04 / 0.35 | 0.722 / 0.628 | 11.60 / 54.19 | 1.07 / 0.45 | 0.536 / 0.439 | 13.03 / 66.14 | 1.00 / 0.31 | 0.561 / 0.478 | 12.43 / 41.16 | 1.10 / 0.55 |
| MPRNet | 11.35 / 46.26 | 1.02 / 0.47 | 0.697 / 0.699 | 15.64 / 42.56 | 0.91 / 0.51 | 0.569 / 0.499 | 14.68 / 51.79 | 1.00 / 0.41 | 0.504 / 0.501 | 12.43 / 41.16 | 0.98 / 0.52 |
| Crop | 14.38 / 19.30 | 0.93 / 0.77 | 0.740 / 0.575 | 13.56 / 18.29 | 1.02 / 0.79 | 0.452 / 0.310 | 17.05 / 21.35 | 0.90 / 0.76 | 0.576 / 0.409 | 15.06 / 14.44 | 0.97 / 0.90 |
| Random Noise | 16.43 / 47.75 | 0.83 / 0.46 | 0.727 / 0.781 | 14.73 / 46.72 | 0.97 / 0.51 | 0.473 / 0.435 | 14.89 / 52.67 | 0.90 / 0.40 | 0.498 / 0.566 | 16.26 / 46.33 | 0.84 / 0.54 |
| Flip | 8.89 / 74.64 | 1.16 / 0.31 | 0.762 / 0.774 | 7.41 / 67.97 | 1.28 / 0.38 | 0.477 / 0.431 | 9.46 / 81.95 | 1.16 / 0.26 | 0.556 / 0.558 | 7.64 / 54.15 | 1.20 / 0.53 |
| Rotate | 10.29 / 17.72 | 1.09 / 0.84 | 0.696 / 0.749 | 9.97 / 15.34 | 1.13 / 0.91 | 0.452 / 0.447 | 12.08 / 22.70 | 1.05 / 0.74 | 0.549 / 0.560 | 9.37 / 13.97 | 1.10 / 0.94 |
| DISCO | 7.77 / 54.65 | 1.20 / 0.38 | 0.720 / 0.735 | 9.10 / 52.58 | 1.17 / 0.45 | 0.522 / 0.459 | 11.26 / 61.65 | 1.05 / 0.36 | 0.543 / 0.542 | 11.28 / 41.30 | 1.15 / 0.61 |
| DiffPure | 17.53 / 23.15 | 0.76 / 0.73 | 0.550 / 0.554 | 19.29 / 22.54 | 0.77 / 0.77 | 0.522 / 0.477 | 17.67 / 21.91 | 0.83 / 0.76 | 0.463 / 0.436 | 12.35 / 20.82 | 0.89 / 0.78 |

*Table 20.* Comparison of $SROCC$ and $PLCC$ scores averaged across KonIQ, KADID and AGIQA-3K datasets for purification-based and adversarial training defenses.

| Defense | Common | | Non-adaptive case | | Adaptive case | |
|---|---|---|---|---|---|---|
| | $SROCC_{clear}\uparrow$ | $PLCC_{clear}\uparrow$ | $PLCC_{adv}\uparrow$ | $SROCC_{adv}\uparrow$ | $SROCC_{adv}\uparrow$ | $PLCC_{adv}\uparrow$ |
| W/o Defense | 0.617±0.01 | 0.648±0.02 | 0.484±0.13 | 0.464±0.12 | 0.402±0.08 | 0.432±0.08 |
| Unsharp | 0.604±0.02 | 0.631±0.02 | 0.452±0.14 | 0.433±0.14 | 0.345±0.12 | 0.366±0.12 |
| Color Quantization | 0.594±0.02 | 0.616±0.02 | 0.574±0.09 | 0.542±0.09 | — | — |
| FCN | 0.580±0.02 | 0.591±0.01 | 0.522±0.10 | 0.498±0.10 | 0.282±0.13 | 0.299±0.12 |
| Bilinear Upscale | 0.577±0.02 | 0.614±0.03 | 0.499±0.12 | 0.468±0.11 | 0.347±0.09 | 0.376±0.09 |
| Gaussian Blur | 0.543±0.03 | 0.572±0.03 | 0.450±0.13 | 0.426±0.12 | 0.360±0.11 | 0.390±0.10 |
| Median Blur | 0.546±0.02 | 0.579±0.02 | 0.458±0.11 | 0.430±0.11 | 0.412±0.11 | 0.444±0.11 |
| JPEG | 0.630±0.02 | 0.655±0.02 | 0.637±0.05 | 0.610±0.04 | — | — |
| DiffJPEG | **0.632±0.02** | **0.658±0.02** | **0.639±0.04** | **0.613±0.04** | **0.548±0.07** | **0.584±0.06** |
| MPRNet | 0.560±0.03 | 0.595±0.04 | 0.570±0.06 | 0.533±0.05 | 0.483±0.08 | 0.513±0.08 |
| Crop | 0.589±0.01 | 0.624±0.02 | 0.553±0.08 | 0.518±0.07 | 0.381±0.10 | 0.389±0.09 |
| Random Noise | 0.566±0.03 | 0.593±0.04 | 0.573±0.05 | 0.547±0.05 | 0.501±0.12 | 0.534±0.11 |
| Resize | 0.606±0.02 | 0.640±0.03 | 0.588±0.07 | 0.561±0.06 | 0.335±0.10 | 0.370±0.10 |
| Real-ESRGAN | 0.570±0.05 | 0.585±0.04 | 0.537±0.08 | 0.523±0.09 | 0.435±0.08 | 0.467±0.07 |
| Flip | 0.598±0.01 | 0.631±0.02 | 0.597±0.05 | 0.564±0.05 | 0.403±0.12 | 0.423±0.11 |
| Rotate | 0.566±0.01 | 0.595±0.02 | 0.530±0.07 | 0.511±0.06 | 0.459±0.10 | 0.488±0.09 |
| DISCO | 0.595±0.02 | 0.626±0.03 | 0.617±0.05 | 0.589±0.03 | 0.466±0.05 | 0.494±0.05 |
| DiffPure | 0.512±0.04 | 0.547±0.04 | 0.531±0.06 | 0.497±0.06 | 0.472±0.04 | 0.511±0.04 |
| APGD-LPIPS-2 | 0.642±0.00 | — | — | — | 0.510±0.09 | 0.449±0.12 |
| APGD-LPIPS-4 | 0.669±0.00 | — | — | — | 0.485±0.14 | 0.475±0.16 |
| APGD-LPIPS-8 | 0.663±0.00 | — | — | — | 0.461±0.11 | 0.420±0.12 |
| APGD-SSIM-2 | 0.620±0.00 | — | — | — | 0.586±0.02 | 0.504±0.07 |
| APGD-SSIM-4 | 0.670±0.00 | — | — | — | 0.445±0.14 | 0.428±0.17 |
| APGD-SSIM-8 | 0.675±0.00 | — | — | — | 0.504±0.10 | 0.509±0.12 |

*Table 21.* Comparison of guarantees and computational complexity of certified defenses. For classification-based methods (C) we measured certified radius and number of abstains, for regression-based methods (R) – certified relative delta.

| Defense | $Cert.R\uparrow$ for **C** $Cert.RD\uparrow$ for **R** | $Abst.\downarrow,\%$ | $Time.\downarrow$, sec |
|---|---|---|---|
| Random. Smoothing (RS) **(C)** | **0.20±0.03** / 0.16±0.04 | **4.68±2.85** / 10.61±5.51 | 36.70±18.68 |
| Denoised RS **(C)** | 0.19±0.03 / 0.16±0.04 | 6.30±3.42 / 7.43±4.51 | 44.43±20.97 |
| Diffusion DRS **(C)** | 0.18±0.02 / 0.17±0.03 | 10.55±5.50 / 8.33±5.47 | 71.85±18.94 |
| DensePure **(C)** | 0.17±0.02 / 0.16±0.02 | 7.91±3.95 / 9.21±6.81 | 116.37±19.04 |
| Median Smoothing (MS) **(R)** | **1.47±0.42** / 2.25±0.79 | - | **26.08±17.05** |
| Denoised MS **(R)** | 1.41±0.34 / 1.88±0.49 | - | 29.35±17.06 |

