# OpenReview forum: "Guardians of Image Quality: Benchmarking Defenses Against Adversarial Attacks on Image Quality Metrics"
_ICML.cc/2025/Conference — ICML 2025 poster_

### Official Review · Reviewer_Dmx9 · 2025-03-11

**Overall Recommendation:** 3

**Summary:**

This paper presents the first comprehensive benchmark study on defense mechanisms for image quality assessment (IQA) metrics, systematically evaluating the performance of 30 defense strategies against 14 adversarial attacks on 9 IQA models.

**Claims And Evidence:**

All the content submitted has corresponding evidence to support it.

**Essential References Not Discussed:**

The author has cited the vast majority of relevant literature. The third point of weakness is the need to supplement additional reference paper.

**Ethical Review Flag:**

Flag this paper for an ethics review.

**Experimental Designs Or Analyses:**

The paper provides ample experiments, which can largely substantiate the authors' viewpoints.The author provides abundant experimental results that strongly support their views.

**Methods And Evaluation Criteria:**

The methods and evaluation criteria proposed have great novelty and pioneering significance.

**Other Comments Or Suggestions:**

My suggestions are already included in the weaknesses discussed earlier.

**Other Strengths And Weaknesses:**

Strengths:
1. The paper has a clear structure and rich content, with valuable experiments provided in the appendix.
2. The author has introduced large-scale subjective studies, and the substantial feedback results further validate the effectiveness of the defense mechanism.

Weaknesses:

1. The paper states that diffusion defense is effective in classification tasks but performs poorly in IQA tasks. However, it seems to lack in-depth analysis, merely mentioning that “future research needs task-specific adjustments.”
2. In Table 2, the time used by DiffPure is 691.42 seconds. However, it appears to only be theoretically feasible and difficult to apply in real-time.
3. Can the authors highlight the significant differences between this work and the existing work [1]? A more detailed response from the authors would better help me resolve my confusion and evaluate this paper.
4. Have existing attack methods considered targeted attacks (e.g., disguising low-quality images as specific high-scoring ones)? Such methods would have a greater impact on ranking.

[1] Kovalev E, Bychkov G, Abud K, et al. Exploring adversarial robustness of JPEG AI: methodology, comparison and new methods[J]. arXiv preprint arXiv:2411.11795, 2024.

**Questions For Authors:**

Q1. In Table 3, why is the training time provided for Cert Defense but not for AT Defense? Q2. When forcibly discretizing regression-based metrics into classification problems, have you considered the potential bias that may arise? This could potentially affect the defense performance. Q3. The defense results show significant fluctuations on the AGIQA-3K dataset, but the authors seem to have not conducted an in-depth analysis of the impact of data bias.  Q4.If the issues are clarified, I will consider increasing the score.

**Relation To Broader Scientific Literature:**

The author presents an interesting application prospect in the field of quality evaluation, using a defense model to assess IQA.

**Theoretical Claims:**

The paper lacks substantial theoretical proof.

---

> ### Author Rebuttal · Authors · 2025-03-31
>
> We sincerely thank the reviewers for their thorough evaluation and constructive feedback, and will apply their recommendations to refine the revised paper. We answer your questions below:
> 1. For adversarial training, there is no inference overhead since the fine-tuning is done during the training phase. In contrast, purification and certified defenses require additional computation during inference, which is why their training time is explicitly reported.
> 2. Yes, we conducted additional experiments to identify the optimal number of classes for discretization. The main challenge is balancing the trade-off between correlations SROCC and certified radius R. As the number of classes increases, SROCC increases, but R decreases. This occurs because a higher number of classes makes it easier to cross class borders during Monte Carlo sampling. The results of these experiments can be found in Table 9, located in Appendix A.3.4.
> 3. Our analysis does not indicate significant fluctuations on the AGIQA-3K dataset specifically. The differences in ranking across datasets are minor and observed on all datasets; the closest one to AGIQA-3K is KADID1K. While data bias is important, our results suggest no substantial difference between AGIQA-3K and natural image datasets. Further investigation into potential biases remains a promising direction for future work.
>
> We address your concerns below:
> 1. In classification tasks, diffusion-based defenses only need to generate images that preserve class-relevant features, making perceptual quality less critical. In contrast, IQA tasks require preserving fine image details while removing adversarial noise, which standard diffusion models struggle with. To improve their applicability, future research should focus on task-specific adjustments, such as incorporating perceptual constraints or adapting diffusion models to minimize distortions introduced during defense.
> 2. We report time complexity in milliseconds, so DiffPure processes an image in ~0.7 seconds. This makes it applicable in real-world applications.
> 3. The mentioned study focuses on the robustness of neural-based image codecs, which makes two significant differences with our work: the area of research in our paper are IQA models, and instead of focusing on the robustness of the models themself, we focus on the effectiveness of defenses. Compared to [1] we use significantly different methodology, including datasets, evaluation metrics and a subjective study. Moreover, we evaluate 30 defense methods of three types (adversarial training, purification, and certified methods), while the authors of [1] used only 7 purification defenses.
> 4. Indeed, we considered exactly this kind of attack. The formal definition is in section 3.1, page 2.
>
> [1] Kovalev E, Bychkov G, Abud K, et al. Exploring adversarial robustness of JPEG AI: methodology, comparison and new methods[J]. arXiv preprint arXiv:2411.11795, 2024.
>
> If you have no further concerns, we would be sincerely grateful if you could consider raising the rating of our submission.

---

### Official Review · Reviewer_Dhjc · 2025-03-11

**Overall Recommendation:** 3

**Summary:**

This paper addresses the vulnerability of neural-network-based Image Quality Assessment (IQA) metrics to adversarial attacks. The authors present the first comprehensive benchmark for evaluating defense mechanisms against such attacks. The study systematically evaluates 30 defense strategies—including purification, training-based, and certified methods—against 14 adversarial attacks in both adaptive and non-adaptive settings across 9 no-reference IQA metrics. The authors also conduct an extensive subjective study with over 60,000 responses to evaluate the perceptual quality of defended images. The benchmark aims to guide the development of more robust IQA defense methods and is open to submissions.

## update after rebuttal
Thanks for the authors' reply. They addressed my questions. After referring to the opinions of other reviewers, I decide to keep my score.

**Claims And Evidence:**

The authors make several claims that are generally well-supported by their evidence:

1. The claim that neural-network-based IQA metrics are vulnerable to adversarial attacks is established through references to prior work and demonstrated in their experiments. The evidence provided is convincing.
2. The claim that this is the first comprehensive benchmark for IQA defense evaluation seems accurate based on the literature review provided on page 2, where they note that existing comparisons focus primarily on object classification rather than IQA metrics.
3. The effectiveness of various defense methods is substantiated through extensive experiments across multiple attack types, datasets, and defense strategies. The authors provide robust statistical analysis to support their conclusions.
4. The authors' observation that compression-based defenses (DiffJPEG, JPEG) are particularly effective is supported by both objective metrics and subjective evaluation.
5. The claim that different defense methods work better for different IQA model architectures (page 7-8) is well-supported by the analyses in Figure 4.
A minor limitation is that while the authors emphasize the importance of their subjective study for evaluating perceptual quality, they don't fully integrate these findings into their main recommendations. Figure 5 (page 13) shows that Real-ESRGAN performs best in subjective evaluation, yet this finding isn't prominently emphasized in the conclusion.

**Essential References Not Discussed:**

no, the citation is appropriate

**Experimental Designs Or Analyses:**

The experimental design is comprehensive and thorough:
1. The authors evaluate defenses across multiple dimensions: effectiveness against different attacks (Table 14), performance on different IQA models (Tables 12-13), and behavior on different datasets (Table 19).
2. The statistical significance testing using Wilcoxon Signed Rank Test with Bonferroni correction (Page 18-19, section A.4) strengthens the validity of their comparisons.
4. The analysis of defense performance across different attack strengths (Table 11) provides valuable insights about defense robustness.

A few concerns about the experimental design:
1. In page 5, section 3.6, the authors mention using 40 NVIDIA Tesla A100 GPUs, but don't specify how computational constraints influenced their experimental decisions, such as limiting certified defense evaluations to only 10 images.
2. The authors don't fully explore potential trade-offs between defense effectiveness and inference time for real-time applications. While they report computation times (Tables 2 and 3), the implications for practical deployment aren't thoroughly discussed.

**Methods And Evaluation Criteria:**

The methodology is generally sound and appropriate for the problem:
1. The authors' approach of evaluating across multiple datasets (Page 4, section 3.4) is commendable, as it ensures results are not dataset-specific.
2. The attack parameter selection method (Page 14-15, section A.3.4) is thoughtfully designed, ensuring fair comparison across attack types by aligning them to target "weak", "medium", and "strong" perturbation levels.
3. The evaluation metrics (Page 4-5, section 3.5) are well-chosen to capture different aspects of defense effectiveness (robustness, perceptual quality preservation, correlation with human judgment).
4. The subjective study design (Page 12, section A.2) using the Bradley-Terry model for pairwise comparisons is appropriate for assessing perceptual quality.

However, I found some methodological concerns:
1. Page 13, Table 5 shows validation of the sampling methodology, but p-values of 0.1449 and 0.1958 suggest the differences between samples might be approaching significance (if using α=0.05). This deserves more discussion.
2. In the evaluation of adversarial training (Page 15-16, section A.3.5), the authors acknowledge that using ground-truth labels from clean images is inaccurate for adversarial examples, yet they don't fully address how this limitation affects their conclusions about adversarial training methods.

**Other Comments Or Suggestions:**

1. Page 2, paragraph 1: The authors state, "making the development of a universally efficient and robust IQA metric impractical." This claim would benefit from more substantiation, as it's a significant assertion.
2. Page 2, paragraph 1: The sentence "To address this, we present a systematic comparison..." seems to imply that their work addresses the impracticality of creating universally robust metrics, but instead it focuses on enhancing existing models through defense mechanisms. This connection should be clarified.
3. Figure 1 provides a good overview, but the explanation of how the components interact could be improved.
4. Page 20, lines 1230-1231: The sentence "In summary, results do not change much across strength" contradicts some of the data in Table 11, where performance differences between weak and strong attacks are substantial for some methods.
5. The paper would benefit from a clearer discussion of the practical deployment considerations for these defense methods in real-world applications.

**Other Strengths And Weaknesses:**

Strengths:
1. The paper addresses a significant practical problem, as vulnerabilities in IQA metrics could affect search engine rankings, benchmarking, and content quality assessment.
2. The benchmark methodology is thoroughly documented and reproducible, with implementation details and code availability mentioned.
3. The large-scale subjective study (60,000+ responses) adds significant value by assessing perceptual quality, which cannot be fully captured by objective metrics.
4. The paper provides nuanced insights, such as the observation that transformer-based IQA metrics have greater intrinsic robustness (page 7, paragraph 2).

Weaknesses:
1. The paper occasionally lacks clarity in explaining the practical implications of its findings. For example, the discussion of certified methods (page 6-7) focuses on technical metrics without clearly articulating the trade-offs involved in deploying such methods.
2. In paragraph 2 of page 3, the authors state they increase IQA scores during attacks "to reflect real-life applications," but don't adequately explain why this is more realistic than decreasing scores.
3. The limitations section in the appendix (page 12) feels somewhat disconnected from the main paper. Some of these limitations, particularly regarding attack parameter handling, should be integrated into the main discussion.
4. The paper introduces many evaluation metrics, which occasionally makes it difficult to determine which metrics should be prioritized when comparing defenses.

**Questions For Authors:**

1. In your subjective study, Real-ESRGAN emerged as the top-performing method for perceptual quality (page 8), yet it doesn't perform as well on objective metrics. How do you reconcile this discrepancy, and what implications does this have for developing or selecting defense methods in real-world applications where human perception is the ultimate measure of quality?
2. Your findings show that transformer-based metrics demonstrate greater intrinsic robustness against adversarial attacks (page 7). Could you elaborate on the architectural features that might contribute to this robustness, and do you believe future IQA metrics should prioritize transformer-based architectures specifically for their robustness properties?
3. You mention on page 6 that certified methods are highly impractical due to computational overhead. Given the trade-off between security guarantees and computational efficiency, what modifications to certified methods might make them more viable for real-world IQA applications?
4. Figure 3(b) suggests some defense methods perform differently on AI-generated images versus natural images. As AI-generated content becomes increasingly prevalent, how might defense strategies need to evolve specifically to address the unique characteristics of such content?

**Relation To Broader Scientific Literature:**

The authors position their work well within the broader literature:
1. They acknowledge prior work on adversarial attacks against IQA metrics (page 1, paragraph 2) while noting the gap in defense mechanism research.
2. They correctly observe that most defense methods have been developed for object classification rather than IQA tasks (page 2, paragraph 1).
3. The authors appropriately reference relevant work on adversarial training, purification methods, and certified defenses (page 2).

One area where the connection to broader literature could be strengthened is in relating their findings to the general principles of adversarial robustness. The paper focuses on the specific domain of IQA metrics, but could better articulate how their findings might generalize to other regression-based neural network tasks.

**Theoretical Claims:**

The paper is primarily empirical rather than theoretical. The authors reference theoretical guarantees for certified defense methods (page 6, page 7) but do not develop new theoretical results of their own.

The authors correctly note on page 7 that "Despite the questionable applicability of randomized smoothing-based defenses to IQA, they remain effective, as the certified radii are sufficiently high and the number of abstentions is relatively low." This is an important observation about the practical utility of theoretical guarantees in this context.

---

> ### Author Rebuttal · Authors · 2025-03-31
>
> Thank you for your valuable suggestions and thoughtful feedback. We will use suggestions to enhance the revised version of the paper. Your questions are answered below:
> 1. The discrepancy between subjective and objective metrics is not uncommon in IQA tasks. While Real-ESRGAN may not achieve the best scores on metrics like PSNR or SSIM, its superior performance in subjective study indicates that it preserves natural image characteristics. In practical applications, these findings imply that defense methods should be evaluated using both objective and subjective measures. Ultimately, this balance is crucial for developing defenses that improve numerical performance and align with how humans perceive images.
> 2. Transformer-based IQA metrics likely benefit from their global attention mechanisms, which enable them to capture long-range dependencies and contextual relationships. Also, pretraining on large datasets can contribute to their generalization capabilities and robustness. While transformer-based architectures offer robustness advantages, they also have increased computational costs. Future IQA research may explore hybrid approaches that combine the strengths of transformers and CNNs, rather than solely prioritizing one architecture based on robustness alone.
> 3. Certified methods provide strong security guarantees but are extremely slow because of their design. To optimize them, one could leverage more efficient certification techniques balancing the trade-off between guarantee and number of samples, or incorporate lightweight architectural modifications. Additionally, hybrid approaches that combine certified defenses with empirical ones may offer a balance between robustness and efficiency. Our benchmark provides a foundation for assessing such trade-offs and guiding future improvements.
> 4. According to Figure 3, the performance on the AI-generated AGIQA dataset is very similar to the performance on the natural KONIQ dataset for most of the defenses, so we disagree that there is a difference in the results. We suggest general ML practices, covering all possible cases during training, including both types of content, to help defenses to generalize better. However, our results show no significant difference in defense efficiency between natural and AI-generated content.
>
> We address your concerns below:
> 1. Statistical tests cannot prove two samples are identical; a p-value above the alpha level means we cannot reject the null hypothesis. In our case, the p-values combined with the small variance of means reported in Table 5, indicate a high level of consistency in our sampling methodology. Figure 6 supports this, showing that score distributions across samples are nearly identical, with closely aligned means. Together, these results demonstrate that any observed differences are statistically negligible and do not compromise the validity of our sampling approach.
> 2. Our study compares two existing solutions for adversarial training of IQA metrics: in one of them, MOS values for adversarial images are penalized, and in another, original labels are used for training on clean data. We highlighted that in general, adversarial training methods that use ground-truth labels for adversarial images may yield decreased correlations, which is not a limitation of our procedure, but encourages future defenses to consider this issue
> 3. Computation time grows dramatically with the number of source images, IQA models and attack methods. Thus, using only 10 source images, we need to apply each certified defense method 1260 times. Overall, we spent about 3000 GPU-hours on certified defenses and ~25000 GPU-hours on all calculations.
> 4. There is a distinct difference in inference time across defense types. For adversarial training, there is no inference overhead since the fine-tuning is done during training. For purification defenses, most methods add only about 0.05–0.2 s per image. This overhead is generally manageable for real-time cases, e.g. quality screening for CCTV, video streaming. Certified defenses offer strong theoretical guarantees but are significantly slower than other methods. This trade-off makes them most suitable for highly sensitive applications where robustness is a necessity, e.g. in benchmarks.
> 5. Our reported computation times in Tables 2, 3 enable developers to choose a suitable defense method based on their deployment constraints. In conclusion, for real-time practical deployment adversarial training and fast purification methods are the best fit.
> 6. Practical applications usually seek to inflate the quality of content. For example, cheating in benchmarks that influence project investments, increasing bitrate after transcoding in target network, inflating results (e.g., Google’s libaom encoder with its --tune-vmaf option). Decreasing scores don't lead to these problems because most real-world systems are aiming to higher quality. Given these scenarios, we prioritized score increases in our approach.

---

### Official Review · Reviewer_ZZxf · 2025-03-16

**Overall Recommendation:** 4

**Summary:**

The manuscript presents a comprehensive benchmark on defenses against adversarial attacks targeting neural network-based Image Quality Assessment (IQA) metrics. It evaluates 30 defense strategies across three categories (purification, adversarial training, and certified methods) against 14 adversarial attacks. The study covers adaptive and non-adaptive attacks on nine no-reference IQA metrics, with extensive experimentation on multiple datasets. The paper introduces a benchmark, a novel dataset of adversarial images, and an online leaderboard, aiming to guide future research in robust IQA metric development.

## update after rebuttal
The positive judgment, also shared by the other reviewers, on this manuscript is confirmed having seen the responses to my comments.

**Claims And Evidence:**

- They argue that defenses such as compression-based methods are particularly effective against adversarial noise, and the experimental results (e.g., DiffJPEG outperforming other purification methods) substantiate this.
- The manuscript asserts that transformer-based IQA models are naturally more robust than CNN-based ones; the reported robustness scores and attack performance trends align with this claim.
- The claim that randomized smoothing methods provide theoretical guarantees is valid but could be expanded with more discussion on its practical applicability to IQA.

**Essential References Not Discussed:**

While the manuscript is well-referenced, some recent works on adversarial robustness in vision models beyond IQA (e.g., adversarial defenses for GAN-based image generation) could provide additional context.

**Experimental Designs Or Analyses:**

- The experimental design is robust, with a diverse dataset selection and comprehensive attack-defense comparisons.
- The use of adaptive attacks strengthens the validity of the results.
- The subjective evaluation study (60,000+ responses) adds significant value but lacks a detailed breakdown of the participant demographics and potential biases.
- A minor concern is that some defenses (e.g., adversarial training) were only evaluated on select IQA metrics, which could impact generalizability.

**Methods And Evaluation Criteria:**

- The methodology is thorough, covering multiple attack strengths, defense configurations, and datasets.
- The evaluation is rigorous, using a mix of objective robustness metrics (e.g., R_{score} and D_{score}) and perceptual quality measures (PSNR, SSIM, crowd-sourced MOS study).
- Adaptive and non-adaptive attack scenarios are considered, adding realism to the evaluation.
- The computational complexity analysis of defenses is a strength but could be extended with a discussion on real-time applicability in practical IQA settings.

**Other Comments Or Suggestions:**

- The manuscript could discuss whether findings extend to full-reference or reduced-reference IQA models.
- The impact statement could be expanded to address ethical considerations of adversarial robustness in applications like image forensics or content moderation.

**Other Strengths And Weaknesses:**

Strenghts
- First systematic study of defenses for adversarial IQA attacks.
- Extensive experimentation across diverse datasets and metrics.
- Well-defined benchmark with clear evaluation criteria.
- Subjective quality assessment adds real-world validity.
- Open-source dataset and leaderboard promote reproducibility.

Weaknesses
- Computational complexity of defenses is high, limiting real-world deployment discussion.
- Some defenses were only tested on select IQA metrics.
- More details needed on subjective study methodology.

**Questions For Authors:**

1. How were the attack strength levels chosen, and do they align with real-world adversarial scenarios?
2. How do the authors envision practical deployment of the best-performing defenses given their computational costs?
3. Would the inclusion of hybrid defense strategies (e.g., combining purification and adversarial training) improve robustness further?
4. Have the authors considered analyzing transferability of adversarial attacks across different IQA metrics?
5. What steps were taken to ensure diversity and reliability in the subjective study participants?

**Relation To Broader Scientific Literature:**

- The study fills an important gap in adversarial robustness research for IQA.
- The comparison with adversarial robustness benchmarks in classification tasks is insightful.
- It would be beneficial to discuss potential implications for other domains using perceptual metrics, such as medical imaging or deepfake detection.

**Theoretical Claims:**

- The authors correctly differentiate empirical and certified defense methods. However, the manuscript could benefit from a more detailed theoretical discussion on why certified methods are less practical for IQA, given their computational overhead.
- The claim that task-specific adaptations are needed for diffusion-based defenses is reasonable and supported by empirical evidence.
- Some theoretical claims regarding robustness improvements from adversarial training could be better justified.

---

> ### Author Rebuttal · Authors · 2025-03-31
>
> We would like to thank the reviewer for the valuable comments and questions. We appreciate the recognition of our study and analysis and will address the questions below:
> 1. We employed three metrics for attack strength estimation depending on the attack type. For $L_{\infty}$, we chose $\frac{2}{255}$, $\frac{4}{255}$, and $\frac{8}{255}$ as the most common thresholds in similar studies [1,2] and aligned them with potential real-world cases where imperceptibility is critical.
> For perceptual metrics-based attacks, we used SSIM and PSNR to quantify attack strength. To ensure alignment across different attacks, we computed average SSIM and PSNR values on a subset of 1000 images from KonIQ-10k corresponding to the selected $L_{\infty}$ thresholds. For each selected $L_{\infty}$ threshold, we calculate corresponding PSNR and SSIM values. For example, the average SSIM value for $L_{\infty}=\frac{2}{255}$ is approximately 0.9.
>
> [1] Croce, F., et al. "RobustBench: a standardized adversarial robustness benchmark," in NeurIPS 2020, arxiv: abs/2010.09670.
>
> [2] Dong Y. et al. "Benchmarking adversarial robustness on image classification," in proceedings of the IEEE/CVF conference on computer vision and pattern recognition, 2020, С. 321-331
>
> 2. Adversarial training requires no additional cost computation during inference, so this type is suitable for real-time quality measurement scenarios, e.g., image quality screening in CCTV, or for frame-by-frame video quality assessment. Purification methods have a wide range of computational overhead from 6 ms (Real-ESRGAN) to $\sim$140 ms (DISCO), with the slowest being diffusion-based methods ($\sim$700 ms for DiffPure). However, they can still be applied to image hostings to measure the quality of new images. Certified methods are much slower (3 to 40 seconds per image). They are best suited for security-sensitive scenarios where robust guarantees outweigh computational costs ー for example, in benchmarks to prevent cheating.
> 3. Combining purification and adversarial training could enhance robustness by leveraging their complementary strengths. Purification handles unseen perturbations, while adversarial training improves resilience to known attack patterns. This hybrid strategy may offer stronger robustness against adaptive attacks while minimizing computational costs. However, AT and purification methods decrease the correlation of IQA metrics with subjective quality, so a combined defense may yield a lower correlation.
> In this work, we focus on existing methods to establish a baseline before exploring more complex, combined approaches in future work.
> 4. We did not analyze the transferability of adversarial attacks across IQA metrics as the paper focuses on evaluating defense mechanisms. Some studies focused on attack methods did analyze transferability[1]. This is recognized as an important direction for future work on adversarial attacks to understand cross-model vulnerabilities.
>
> [1] A. Ghildyal, F. Liu, "Attacking Perceptual Similarity Metrics," in Transactions on Machine Learning Research, 2023
>
> 5. To ensure diversity and reliability, we conducted a large-scale crowd-sourced study using the Subjectify.us platform, which enabled us to gather opinions from a broad and diverse participant pool. This platform helps reduce sampling bias and expands the range of participants. There were also participants’ answers quality control measures implemented via verification questions. Furthermore, the platform ensures that each assessor can only participate once and that the total number of answers for each pair is at least 10. This process minimizes the impact of unreliable inputs and ensures the robustness of our subjective evaluations.

---

### Official Review · Reviewer_BpWG · 2025-03-16

**Overall Recommendation:** 3

**Summary:**

The paper proposes a benchmark for defending neural-network based image quality assesment (IQA) against adversarial attacks. The paper makes an extensive study with numerous datasets, IQA models and adversarial attacks of different types and discusses the evaluation results.

**Claims And Evidence:**

I think, the claim of proposing a comprehensive benchmark for evaluating defences against adversarial attacks on image quality assesment is supported in the paper. Up to my knowledge, there were no such benchmarks up to now.

**Essential References Not Discussed:**

I am not aware of any essential references that were not discussed.

**Experimental Designs Or Analyses:**

The validity of experimental designs appears to be adequate, but it was not carefully checked.

**Methods And Evaluation Criteria:**

I think, the datasets, IQA models, adversarial attacks and evaluation metrics make sense for the problem.

**Other Comments Or Suggestions:**

$-$

**Other Strengths And Weaknesses:**

Strengths

1. An extensive defense evaluation including adaptive and non-adaptive adversarial attacks.
2. A large-scale subjective study with many participants to provide additional evaluation.
3. Since the proposed benchmark is claimed to be open for submissions, it may facilitate the development of the field.

Weaknesses

1. I appreciate the effort made in this paper to provide a benchmark for the IQA defences. The main concern is whether the methodological novelty of this paper is sufficient for the main track of the conference where this paper was submitted. The paper proposes some novel metrics for the considered task (e. g. (8), (9)) but in general the novelty seems to be rather limited. I think the proposed benchmark can be valuable for the community. But in my opinion after some adaption and formulating a clear position on the current state of the IQA defences based on the performed studies, the paper would much better fit into the Positon Paper Track of the conference.

2. Apart from the novelty concerns formulated above, I am not fully convinced regarding the significance of the IQA metric defences in general. The paper makes an example based on the image processing and compression competitions where the competitors might exploit the metrics to land higher on the leaderboard. I am not sure whether it is a significant concern for the ML community which requires developing specific defenses for image quality assessment. Thus, the significance of the proposed benchmark is also questionable

**Questions For Authors:**

No further questions.

**Relation To Broader Scientific Literature:**

The paper relies on existing adversarial attacks and previously proposed IQA methods.

**Theoretical Claims:**

I have not seen any theoretical claims or proofs that need to be checked.

---

> ### Author Rebuttal · Authors · 2025-03-31
>
> Thank you for your thoughtful and detailed feedback. We truly value your questions and your time reviewing our paper. We address your concerns below:
> 1. Our contributions include a novel methodology for comparing defenses for the IQA tasks, addressing a critical gap in the field, an extensive subjective study with 60,000+ responses, and an in-depth analysis of the results. Novelty of methodology includes: sampling strategy for dataset, wide range of attacks of various types (including WB and BB, restricted and unrestricted, dadptive and non-adaptive attacks), comparing different types of defenses with each other, aligning attacks parameters by attack strenth, varying defense parameters, new metrics for evaluation and subjective comparison.
> Notably, our work directly aligns with the main topics from the call for papers for ICML 2025 — specifically, Evaluation (Methodology) and Trustworthy Machine Learning (Robustness, Safety). By targeting an underexplored area with clear real-world impact, our benchmark lays the groundwork for future contributions. To our knowledge, no prior work has systematically examined defenses in this context, underscoring the novelty and importance of our approach.
> 2. The manipulation of IQA metrics is a significant concern that extends beyond image processing competitions to numerous critical applications in the ML community. IQA metrics are foundational in detecting copyrighted content, ranking results in internet search engines (e.g., Bing, as noted in Section 1), assessing medical imaging quality for diagnostics [1,2], enhancing face recognition systems[3], and nearly all preprocessing techniques for images and videos.
> The significance of attacking IQA metrics extends beyond manipulating benchmarks and is already recognized by research and industry areas. Existing papers address issues with IQA robustness because IQA is foundational in detecting errors in medical imaging, ranking the results in search engines, etc. Exploring and enhancing the robustness of IQA modes is an active area of research[4,5,6]. Moreover, after the paper[8] about the adversarial vulnerability of VMAF metric and implementation of this attack in Google's libaom video codec, VMAF developers from Netflix had to release a more robust version VMAF NEG[7], highlighting the importance of robust IQA metrics for the industry. We want to establish a standard to evaluate emerging methods and boost the development in this area by publishing our benchmark.
>
> If you have no further concerns we would be sincerely grateful if you could consider raising the rating of our submission.
>
> [1] Yuan S. et al. “A Deep-Learning-Based Label-free No-Reference Image Quality
> Assessment Metric: Application in Sodium MRI Denoising”
>
> [2] Dong X., Fu L., Liu Q. “No-reference image quality assessment for confocal endoscopy images with perceptual local descriptor,” Journal of Biomedical Optics, 2022
>
> [3] Terhorst P. et al. “SER-FIQ: Unsupervised estimation of face image quality based on stochastic embedding robustness,” in Proceedings of the IEEE/CVF conference on computer vision and pattern recognition, 2020
>
> [4] Ghazanfari S. et al. “R-LPIPS: An adversarially robust perceptual similarity metric”
>
> [5] Kettunen M. et al. “E-lpips: robust perceptual image similarity via random transformation ensembles”
>
> [6] Chistyakova A. et al. “Increasing the Robustness of Image Quality Assessment Models Through Adversarial Training”
>
> [7] Netflix blogpost: https://netflixtechblog.com/toward-a-better-quality-metric-for-the-video-community-7ed94e752a30
>
> [8] Zvezdakova A. et al. “Hacking VMAF with Video Color and Contrast Distortion”

---

> > ### Comment · Reviewer_BpWG · 2025-04-02
> >
> > Thank you for addressing the concerns on the novelty and significance of the paper raised in the review. After reading the rebuttal I have decided to raise my score.

---

### Official Review · Reviewer_H2DF · 2025-04-05

**Overall Recommendation:** 3

**Summary:**

Image Quality Assessment (IQA) mostly uses DNNs to calculate the score, leaving space for attackers to perturb the image to manipulate the score for commercial advantage in ranking. Compared to (Antsiferova et al., 2024) that benchmark attacks to IQA, this paper benchmarks defenses in this task. It considers 17 purification (pre-processing) defenses, 2 adversarial training (with variations), and 6 certified defenses. The evaluation is extensively performed on 4 image datasets, 9 IQA models, and 14 attacks (with adaptive variations). Authors publish the benchmark for the community to assess upcoming defenses/attacks.

## update after rebuttal
Thanks the authors for providing the rebuttal in a limited time. I think it is good to dedicate a paragraph to organize existing insights as promised, but overall, more insights from this work are encouraged. I did not question the novelty against RobustBench, but more discussions on the findings on this benchmark and RobustBench should be presented. I decided to keep my score.

**Claims And Evidence:**

Claims are clear.

**Essential References Not Discussed:**

Not found by me.

**Experimental Designs Or Analyses:**

Yes, see Other Strengths And Weaknesses.

**Methods And Evaluation Criteria:**

Yes.

**Other Comments Or Suggestions:**

N/A

**Other Strengths And Weaknesses:**

Strengths
1. Attacks to IQA models are well motivated, contributing to the value and timeliness of a defense benchmark. To achieve a high ranking in a search engine, an image releaser has the motivation to perturb the image to bypass the DNN-based quality ranking system, so that a specific image receives unmatched attention.
2. I am impressed by the comprehensiveness of experiments. For each of the assesed defense, all common attacks are tested. Notably, adaptive attacks that backpropagate attack gradients through the defense are also considered. The authors also study extensively on how the attack/defense hyperparameters affect the results.
3. Various metrics are adopted. Robustness scores reflect how well a defense can restore the IQA scores of adversarial images to match their original values. Quality scores measure the perceptual similarity between purified images and their original images. Performance scores assess an IQA metric’s performance in the presence of adversarial defense.

Weaknesses
1. The takeaways for IQA designers are not clear enough. The benchmark is meant to provide guidance for defenders on how to secure their system. However, only limited discussions are put in Section 5. As a researcher not working in this field, I struggle to learn useful insights from a lot of numbers presented.
2. I am not sure how this work links to previous benchmarks on securing image classifiers, e.g., RobustBench. Both dealing with images, a lot of attacks/defenses are using the same technique. Are there similar conclusions about a defense method? What is new in IQA? There seems to be limited sentences for that, but I think a concentrated paragraph would be helpful. This gives hints on whether it is necessary to develop benchmarks for future new DNN-based image tasks.
3. I believe the manuscript would benefit from presentation improvement in terms of a clear prioritization.

**Questions For Authors:**

As a late reviewer, I do not expect interactions with the authors. Feel free to deprioritize my opinions if rebuttal is important in AC's judgement.

**Relation To Broader Scientific Literature:**

Authors publish the benchmark for the community to assess upcoming IQA defenses/attacks.

**Theoretical Claims:**

No theoretical claims made.

---

### Decision · Program_Chairs · 2025-05-01

**Decision:**

Accept (poster)

**Comment:**

This paper presents a systematic benchmark dedicated to defending neural-network–based Image Quality Assessment (IQA) metrics from adversarial attacks. The evaluation is very comprehensive, studying the effectiveness of 30 defense strategies against 14 attack types on 9 no‐reference IQA metrics. An online leaderboard will also be provided to track the latest progress.

Overall, all reviewers agree that this work is comprehensive and well motivated, the experimental design is thorough, and the analysis yields interesting insights. Meanwhile, some concerns are raised, including 1) the practical guidance for IQA designers is not clear enough; 2) more discussions are needed regarding its relationships to RobustBench; and 3) more in-depth analysis is needed, like computational overhead.

The authors’ rebuttal has satisfactorily addressed most of these issues, leading all reviewers to support the acceptance of this submission.